# Rational design of a JAK1-selective siRNA inhibitor for the modulation of autoimmunity in the skin

Qi Tang [1,2], Hassan H. Fakih [2], Mohammad Zain Ul Abideen[2], Samuel R. Hildebrand[2], Khashayar Afshari[1], Katherine Y. Gross[2], Jacquelyn Sousa[2], Allison S. Maebius[3], Christina Bartholdy[4], Pia Pernille Søgaard [4], Malene Jackerott [4], Vignesh Hariharan [2], Ashley Summers [2], Xueli Fan[1], Ken Okamura [1], Kathryn R. Monopoli[2,5], David A. Cooper [2,3], Dimas Echeverria[2], Brianna Bramato[2], Nicholas McHugh [2], Raymond C. Furgal [2], Karen Dresser [6], Sarah J. Winter[2], Annabelle Biscans [2], Jane Chuprin [1], Nazgol-Sadat Haddadi [1], Shany Sherman[1], Ümmügülsüm Yıldız-Altay [1], Mehdi Rashighi[1], Jillian M. Richmond[1], Claire Bouix-Peter[7], Carine Blanchard [7], Adam Clauss [4], Julia F. Alterman [2] ✉, Anastasia Khvorova [2,3] ✉ & John E. Harris [1] ✉

Inhibition of Janus kinase (JAK) family enzymes is a popular strategy for treating inflammatory and autoimmune skin diseases. In the clinic, small molecule JAK inhibitors show distinct efficacy and safety profiles, likely reflecting variable selectivity for JAK subtypes. Absolute JAK subtype selectivity has not yet been achieved. Here, we rationally design small interfering RNAs (siRNAs) that offer sequence-specific gene silencing of JAK1, narrowing the spectrum of action on JAK-dependent cytokine signaling to maintain efficacy and improve safety. Our fully chemically modified siRNA supports efficient silencing of JAK1 expression in human skin explant and modulation of JAK1-dependent inflammatory signaling. A single injection into mouse skin enables five weeks of duration of effect. In a mouse model of vitiligo, local administration of the JAK1 siRNA significantly reduces skin infiltration of autoreactive CD8+ T cells and prevents epidermal depigmentation. This work establishes a path toward siRNA treatments as a new class of therapeutic modality for inflammatory and autoimmune skin diseases.

Janus kinase (JAK) family enzymes—JAK1, JAK2, JAK3, and tyrosine kinase 2 (TYK2)—transduce signal cascades for a large variety of type I and type II cytokines via the JAK-signal transducer and activator of transcription (STAT) pathway (Fig. 1a)[1]. JAK-dependent cytokines mediate the pathogenesis of many allergic, autoimmune, and inflammatory disorders; therefore, the development of JAK inhibitors has become a popular therapeutic strategy for immunomodulation[2–5]. Indeed, since the historical approval of

[1]Department of Dermatology, University of Massachusetts Chan Medical School, Worcester, MA 01605, USA. [2]RNA Therapeutics Institute, University of Massachusetts Chan Medical School, Worcester, MA 01605, USA. [3]Program in Molecular Medicine, University of Massachusetts Chan Medical School, Worcester, MA 01605, USA. [4]LEO Pharma A/S, Industriparken 55, 2750 Ballerup, Denmark. [5]Bioinformatics and Computational Biology Program, Worcester Polytechnic Institute, Worcester, MA 01609, USA. [6]Department of Pathology, University of Massachusetts Chan Medical School, Worcester, MA 01605, USA. [7]Aldena Therapeutics, London E1 6RA, UK. ✉e-mail: Julia.Alterman@umassmed.edu; Anastasia.Khvorova@umassmed.edu; John.Harris@umassmed.edu

ruxolitinib by the FDA[6] in 2011, JAK inhibitors have dramatically improved the clinical treatment outcomes for rheumatoid arthritis[7,8], inflammatory bowel disease[9–11], and various skin conditions[12–17]. However, most small molecule JAK inhibitors non-covalently bind to the kinase domain, which is relatively conserved across JAK family members[18]. Thus, these drugs commonly inhibit multiple JAKs, and are frequently associated with consequent side effects. Selective, subtype-specific inhibition of JAK to affect a narrow spectrum of action on cytokines may, in principle, improve

safety while retaining efficacy; however, achieving absolute specificity remains challenging.

RNA interference (RNAi) based-therapeutics, such as small interfering RNAs (siRNAs), are a new generation of drug modality expected to dramatically accelerate the advancement of human medicine. siRNAs are double-stranded oligonucleotides that are usually covalently linked to bioconjugates (e.g., ligands, antibodies, and hydrophobic moieties) or formulated with delivery vehicles (e.g., lipid nanoparticles) to improve tissue delivery and cellular uptake. Upon

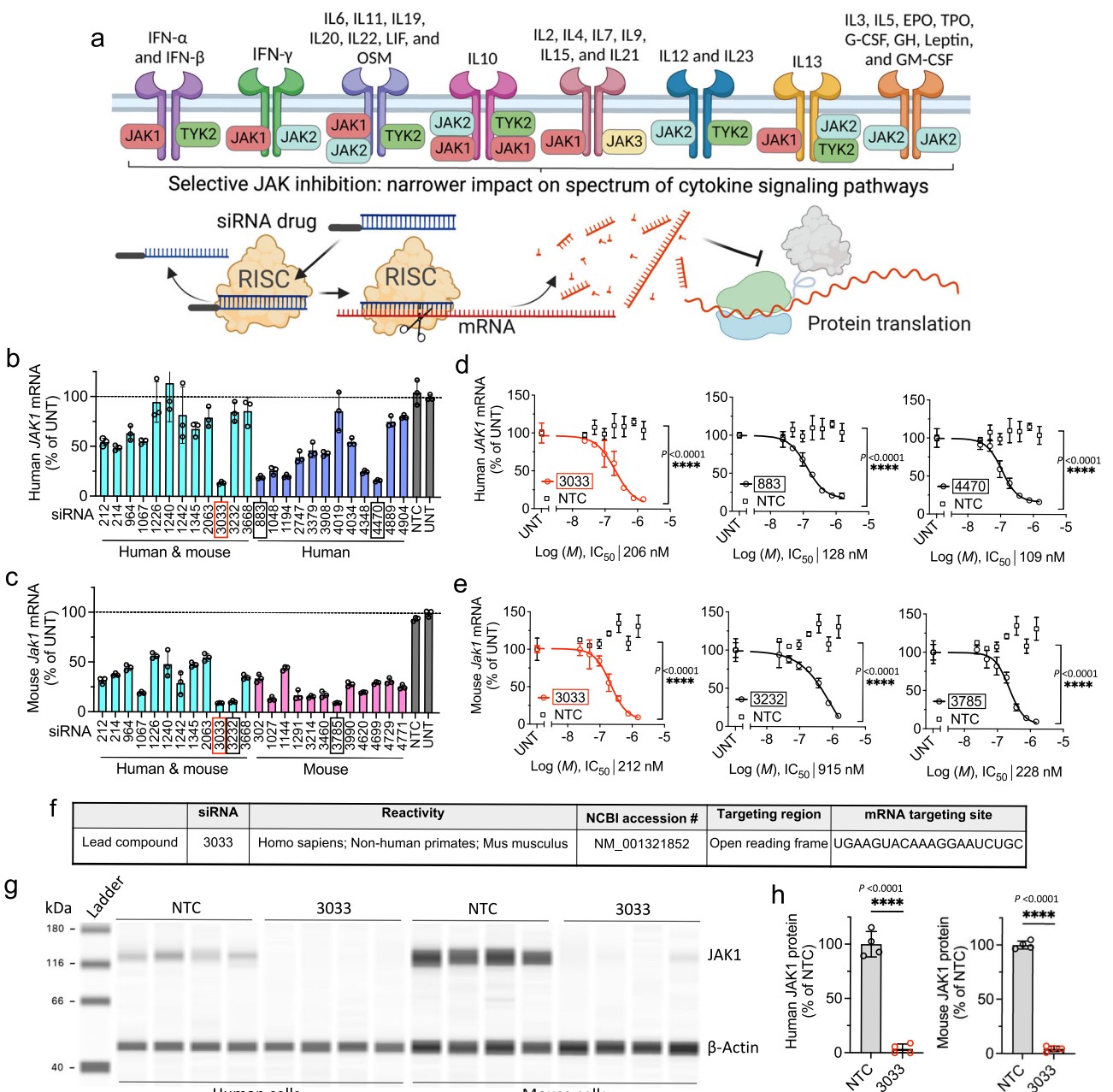

**Fig. 1 | Identification of lead siRNA 3033 that potently silences JAK1 expression.** **a** Mechanism of action of RNAi for selective silencing of JAK1 to narrow down the spectrum of action on inflammatory cytokine pathways. **b, c** In vitro screening of siRNAs in human HeLa (**b**) and mouse N2a (**c**) cells. Cells were treated with fully modified cholesterol-conjugated siRNAs at 1.5 μM for 72 h. mRNA levels were measured by the QuantiGene 2.0 assay. siRNA number represents the 5′-position of the mRNA target site. UNT untreated control, NTC nontargeting control siRNA. Data are represented as percent of UNT (n = 3 biologically independent samples, mean ± s.d.). **d, e** Seven-point dose–response curves of lead siRNAs 3033, 883, 4470

in human HeLa cells (**d**) and lead siRNAs 3033, 3232, and 3785 in mouse N2a cells (**e**). M, molar concentration of siRNA (n = 3 biologically independent samples, mean ± s.d., ****P < 0.0001; two-way ANOVA). **f** Accession number and mRNA targeting site of lead compound 3033. **g** JAK1 protein expression in human HeLa and mouse N2a cells (in a single capillary cartridge using western capillary ProteinSimple assay) at 72 h post siRNA treatment (1.5 μM). **h** Quantification of JAK1 protein in (**g**). (n = 4 biologically independent samples, mean ± s.d. ****P < 0.0001; two-sided unpaired t test). Source data are provided as a Source Data file.

internalization, the antisense strand of the siRNA is loaded into an RNA-induced silencing protein complex (RISC) capable of recognizing and cleaving complementary mRNAs, thus silencing the mRNA and aborting target protein translation (Fig. 1a). Because target specificity is determined simply by the sequence of the antisense strand, and target selection is not limited to accessible extracellular and cell surface molecules (like for antibodies), or requiring iterative screening process (like for small molecules), siRNAs represent a highly programmable molecular entity for selectively silencing a large number of disease targets.

Currently, conjugate-mediated delivery of chemically stabilized siRNAs is the dominant platform in the clinic and is the basis for multiple siRNA drugs approved by the FDA since 2018, covering both rare and common diseases[19–23]. The most advanced siRNA conjugate, N-acetylgalactosamine (GalNAc), supports selective hepatocyte delivery through asialoglycoprotein receptor uptake. GalNAc conjugates exhibit excellent safety and efficacy profiles in humans, with potency and sustainability exceeding the clinical performance of small molecule counterparts[24–26]. More recently, we have shown that hydrophobic conjugates, such as docosanoic acid (DCA), support nontoxic, functional delivery to all major cell types in the skin enabling efficient targeted gene silencing[27]. The adaptability of siRNA therapeutics for new disease targets and the promise of safe in vivo delivery via bioconjugates make siRNAs an ideal therapeutic paradigm for skin diseases.

Here we report the rational discovery and preclinical development of a JAK1-selective siRNA for the modulation of autoimmunity in the skin. We screened a panel of siRNA sequences computationally predicted to silence JAK1 and identified a lead compound, 3033, that is cross-reactive to human, non-human primate, mouse, and rat JAK1 mRNA transcripts. Full chemical stabilization of 3033 enabled efficient JAK1 silencing in human skin (ex vivo) and mouse skin (in vivo), translating to functional inhibition of JAK1-dependent inflammatory signaling. We further demonstrated the therapeutic efficacy of 3033 in a mouse model of autoimmune skin disease (i.e., vitiligo) driven by IFN-γ signaling. Our work reveals that RNAi is a highly programmable approach for designing targeted skin therapies and establishes a path toward siRNA treatment of inflammatory and autoimmune skin diseases.

## Results

### Identification of lead siRNA 3033 that enables potent gene silencing of JAK1

We used our proprietary siRNA design algorithm and generated a panel of 150 siRNA sequences (50 human, 50 mouse, and 50 human/mouse cross-targeting) that are specific to the 5'-untranslated region (UTR), open reading frame, or 3'-UTR regions of JAK1 mRNA transcripts. The top 12 candidates from each category (human, mouse, or human/mouse targeting) were synthesized for in vitro screening. Sequences and chemical modifications of each synthesized compound are summarized in Supplementary Table 1. The 3' end of siRNA sense strands were conjugated to cholesterol to enable efficient in vitro cellular internalization through passive uptake. Primary screening in human HeLa (Fig. 1b) and mouse N2a (Fig. 1c) cells identified several siRNA sequences that enabled 80–90% silencing of human and mouse JAK1 mRNAs, respectively. In both screens, the nontargeting control (NTC) siRNA showed no silencing compared to untreated (UNT) cells. The median inhibition concentration (IC$_{50}$) values were determined for the top three hits from each primary screen (human 3033, 883, and 4470; and mouse 3033, 3232, and 3785) using seven-point dose–response studies (Fig. 1d, e). We selected 3033 as the top candidate for subsequent development as it provided the highest silencing efficacy and potency, and is cross-reactive to human and mouse JAK1 transcripts (Fig. 1f). When we quantified JAK1 protein expression (Wes capillary assay) in the tested human and mouse cell lines, we

observed near-complete silencing of both human and mouse JAK1, indicating 3033 is a highly potent siRNA (Fig. 1g, h). 3033 also demonstrated efficient silencing of non-human primate JAK1 in a *Macaca mulatta* cell line, which is consistent with the significant homology between human and non-human primate JAK1 (Supplementary Fig. 1).

### Selective silencing of JAK1 inhibits IFN-γ signaling

The sequence-specific nature of siRNAs should enable 3033 to selectively silence JAK1, but not other JAK family enzymes. To test this hypothesis, we treated a validated human cell line (HEL 92.1.7) that highly expresses JAK1, JAK2, JAK3, and TYK2 mRNAs with 3033 and measured mRNA levels of each JAK subtype. 3033 significantly silenced JAK1 expression but had no effect on JAK2, JAK3, and TYK2 expression (Fig. 2a). This result suggests that 3033 is strictly selective to JAK1 and the specificity is attributed to the RNAi-based action of mechanism.

We next benchmarked the ability of 3033 to modulate a disease-relevant inflammatory pathway (i.e., IFN-γ-JAK1/2-STAT1-CXCL9/10/11 pathway) against ruxolitinib, a small molecule JAK1/2 inhibitor. Currently, ruxolitinib is the only FDA-approved treatment for improving vitiligo—a CD8+ T-cell-mediated autoimmune skin disease causing melanocyte death and epidermal depigmentation. During disease progression, autoreactive CD8+ T cells in the lesional skin release IFN-γ to activate the JAK1/2-STAT1 pathway in local skin cells[28–30], inducing the expression of chemokines CXCL9, 10, and 11 to promote the infiltration of CD8+ T cells via positive feedback (Fig. 2b). In a human cell-based model of IFN-γ signaling, we treated cells with 3033 or ruxolitinib and then induced IFN-γ signaling using recombinant human IFN-γ protein. 3033 inhibited the expression of CXCL9, 10, and 11 mRNA, showing higher potency than ruxolitinib in vitro (Fig. 2c). However, in vivo delivery of siRNAs versus small molecules into different cell types may vary. Thus, the therapeutic efficacy of 3033 must be evaluated in vivo.

To optimize 3033 for in vivo skin delivery, we covalently linked 3033 to a hydrophobic conjugate, docosanoic acid (DCA)[31], which supports local skin retention and productive delivery of siRNA to all skin cell types in vivo[27]. As the chemical modification pattern of siRNA scaffolds may impact their potency in certain sequence contexts[27,32], we thus compared the potency of a new modification pattern (i.e., scaffold 2) with relatively balanced content of 2'-O-methyl (56%) and 2'-Fluoro (44%) sugar modifications to the modification pattern (scaffold 1; 74% 2'-O-methyl, 26% 2'-Fluoro) used in the initial in vitro experiments (Fig. 2d). The potency of each scaffold was determined in a 7-point dose–response study through lipofectamine-mediated cellular uptake in vitro. We found that scaffold 2 offered a two- to threefold (IC$_{50}$ 12.2 nM vs. 33.1 nM) improvement in potency over scaffold 1 (Fig. 2e). Therefore, scaffold 2 was applied to 3033, creating a compound hereinafter referred to as si3033 (Fig. 2d, f). We next tested the ability of si3033 (versus ruxolitinib) to inhibit IFN-γ signaling in an IFN-γ-JAK1/2-STAT1 luciferase reporter cell line, where luciferase expression is driven by IFN-γ-activated regulatory sites (Fig. 2g). Both si3033 and ruxolitinib inhibited IFN-γ signaling (Fig. 2h, i). As si3033 selectively silences JAK1 mRNA and does not impact the expression of JAK2 (Fig. 2j), this result suggests that selectively targeting JAK1 may be sufficient to inhibit JAK1/2-mediated inflammatory pathways. Our findings support a selective JAK inhibition strategy for reducing the undesired effects of inhibiting multiple JAKs without compromising efficacy.

### si3033 has a good safety profile and enables JAK1 silencing in mouse skin over 5 weeks

We subcutaneously injected wild-type C57Bl/6J mice with 0.4 mg (in 150 μL PBS) of si3033 and measured JAK1 mRNA expression in skin local to the injection site over 5 weeks. si3033 enabled ~40–50%

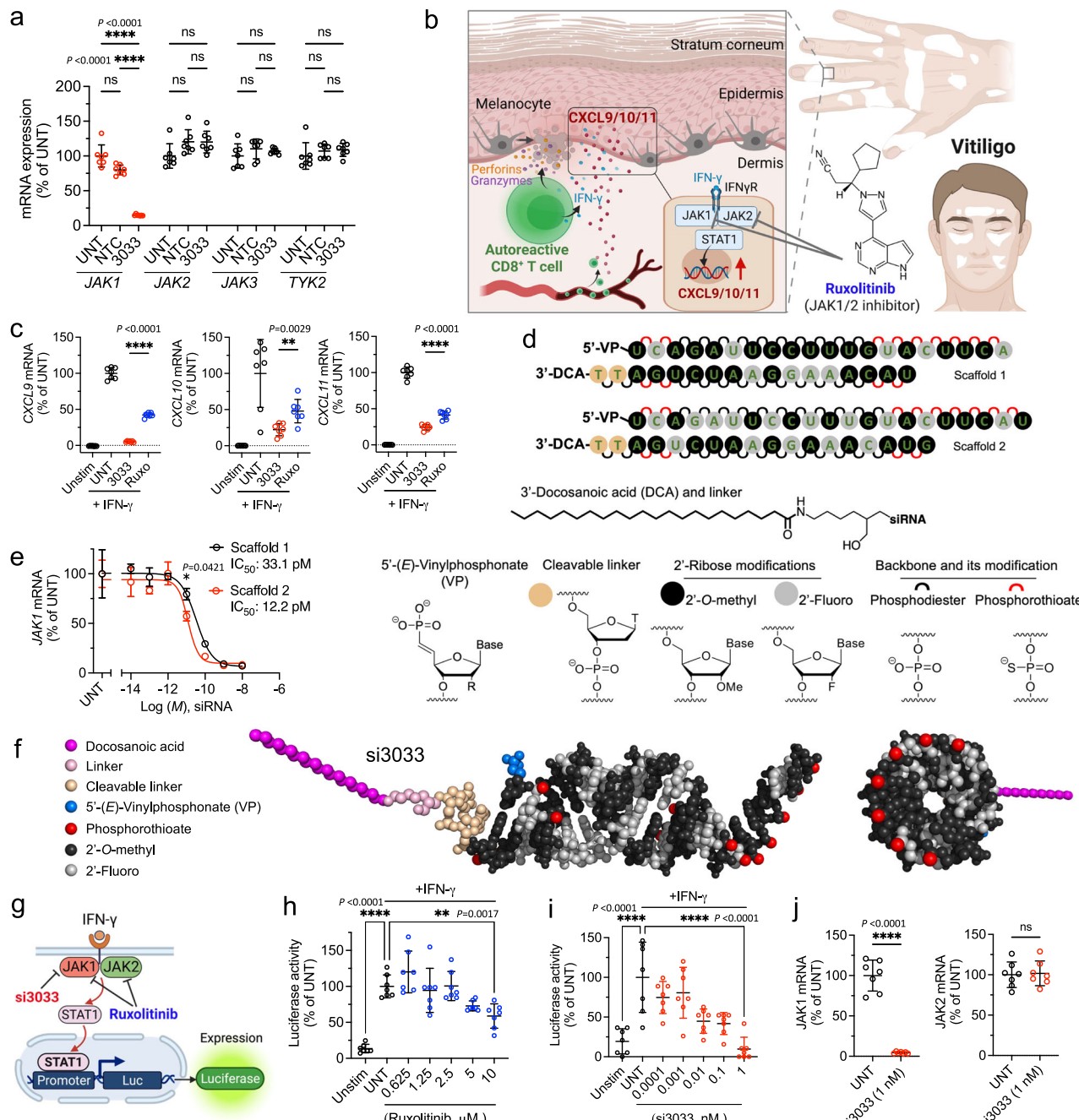

**Fig. 2 | Selective silencing of JAK1 by siRNA inhibits IFN-γ signaling. a** Activity of 3033 on JAK family enzymes. Human HEL 92.1.7 cells were treated with siRNA at 1.5 μM for 72 h, and mRNA levels of JAK1, JAK2, JAK3, and TYK2 were measured using the QuantiGene 2.0 assay ($n = 7$ biologically independent samples, mean ± s.d.; one-way ANOVA, ****$P < 0.0001$, ns: not significant). **b** Ruxolitinib inhibits JAK1 and JAK2 to prevent the pathogenesis of vitiligo driven by IFN-γ signaling. **c** mRNA expression of IFN-γ-inducible chemokines CXCL9, 10, and 11. Cells were first treated with siRNA or ruxolitinib at 1.5 μM for 72 h, and then stimulated with recombinant human IFN-γ for 24 h ($n = 7$ biologically independent samples, mean ± s.d.; one-way ANOVA, **$P < 0.01$, ****$P < 0.0001$). **d, e** Chemical engineering of 3033 for skin delivery. Terminal nucleotides were stabilized with phosphorothioate backbone modifications for nuclease stability (**d**). Human HeLa cells were transfected with DCA-siRNAs for 72 h through lipofectamine RNAiMax-mediated uptake and mRNA levels were measured using the QuantiGene 2.0 assay ($n = 3$ biologically independent samples, mean ± s.d.; two-way ANOVA multiple comparison, *$P < 0.05$). **f** Molecular modeling (PyMOL 2) of DCA-siRNA 3033 in scaffold 2, si3033. **g–i** ruxolitinib and si3033 both reduce luciferase activity in IFN-γ-JAK1/2-STAT1 HeLa reporter cells ($n = 7$ biologically independent samples, mean ± s.d.; one-way ANOVA, **$P < 0.01$, ****$P < 0.0001$). Cells were treated with ruxolitinib or siRNA for 72 h and then stimulated with IFN-γ for 18 h. si3033 was transfected to the HeLa reporter cells using lipofectamine RNAiMax. **j** JAK1 and JAK2 mRNA expression in si3033-treated and untreated HeLa reporter cells ($n = 7$ biologically independent samples, mean ± s.d.; two-sided unpaired $t$ test, ****$P < 0.0001$, ns: not significant). Source data are provided as a Source Data file.

silencing of JAK1 at week 1. Although this level of silencing was not maintained (perhaps due to active renewal of mouse skin), significant downregulation of JAK1 mRNA, compared to siNTC control, lasted for 5 weeks (Fig. 3a). This result indicates that a single injection of si3033 provides at least a 1-month duration of effect in vivo, similar to our previous study on silencing the IFN-γ receptor in skin using DCA-siRNAs[27].

Hydrophobic conjugation of siRNAs to DCA enhances their local retention at the injection site; however, a fraction of the injected dose distributes systemically. To determine the biodistribution of si3033,

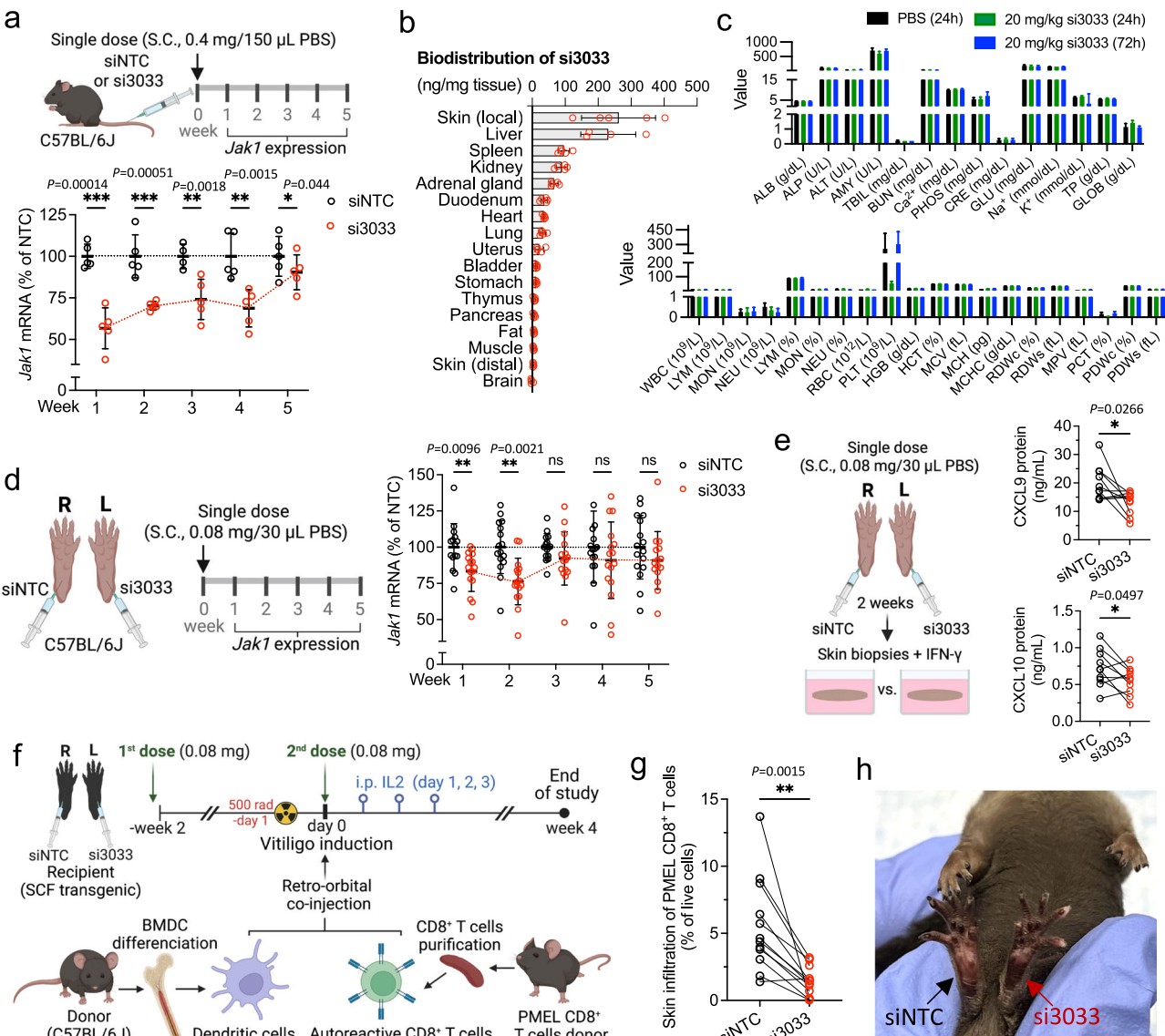

**Fig. 3 | si3033 reduces JAK1-mediated T-cell recruitment to the skin and prevents depigmentation in a mouse model of vitiligo. a** Single-dose subcutaneous (S.C.) injection of si3033 at 20 mg/kg supports Jak1 silencing in tail skin for over 5 weeks ($n = 5$ animals, mean ± s.d.; two-sided unpaired $t$ test, *$P < 0.05$, **$P < 0.01$, ***$P < 0.001$); QuantiGene 2.0 assay and presented as percentage of siNTC control. **b** Biodistribution of si3033 at skin local to the injection site and in systemic tissues. ($n = 5$ animals, mean ± s.d.). **c** Blood chemistry diagnostics at 24 h and 72 h post S.C. injection of 20 mg/kg si3033 ($n = 8$ animals, mean ± s.d. the current plotting is to support visual clarity, raw data values can be found in the supplied source data file). ALB albumin, ALP alkaline phosphatase, ALT alanine transaminase, AMY amylase, $Ca^{2+}$ calcium, CRE creatinine, GLU glucose, $Na^+$ sodium, $K^+$ potassium, TP total protein, GLOB globulin, WBC white blood cell, LYM lymphocyte, MON monocyte, NEU neutrophil, RBC red blood cell, PLT platelet, HGB hemoglobin, HCT hematocrit, MCV mean corpuscular volume, MCH mean corpuscular hemoglobin, MCHC mean corpuscular hemoglobin concentration, RDWc red blood cell distribution width coefficient of variation, RDWs red blood cell distribution width standard deviation, MPV mean platelet volume, PCT procalcitonin, PDWc platelet

distribution width coefficient of variation, PDWs platelet distribution width standard deviation. **d** 0.08 mg of si3033 provides Jak1 silencing in the footpad skin (right vs. left pad) for 2 weeks. Mouse *Jak1* mRNA were measured over 5 weeks using the QuantiGene 2.0 assay ($n = 16$ animals, mean ± s.d.; two-sided paired $t$ test, *$P < 0.05$, **$P < 0.01$, ***$P < 0.001$). **e** si3033 significantly reduces the expression of IFN-γ-inducible chemokines CXCL9 and 10 in an ex vivo skin model of IFN-γ signaling at week 2. IFN-γ signaling was induced using recombinant mouse IFN-γ in a 3-mm skin punch collected at the injection site of footpad. Mouse CXCL9 and CXCL10 were quantified in the ex vivo culture media using ELISA assays ($n = 10$ animals, mean ± s.d.; lines represent two-sided paired $t$ test in the same mouse, *$P < 0.05$). **f** Schematic of a mouse model of vitiligo that mimics human skin depigmentation. **g** Skin infiltration of autoreactive PMEL CD8+ T cells ($n = 12$ animals, mean ± s.d.; two-sided paired $t$ test, **$P < 0.01$). **h** Representative image reveals that si3033 prevents skin depigmentation in footpads. The locally injected area of left footpad exhibited less severity of depigmentation compared to right footpad treated with siNTC control compound. Source data are provided as a Source Data file.

we subcutaneously injected mice with 20 mg/kg of si3033 and determined the amount of siRNA in a total of 17 tissues: skin local to the injection site, liver, spleen, kidney, adrenal gland, duodenum, heart, lung, uterus, bladder, stomach, thymus, pancreas, fat, muscle, skin at distal site, and brain. The antisense strands of si3033 were quantified using a peptide nucleic acid hybridization assay (Supplementary Fig. 2a)[31,33]. We found that si3033 was retained at skin local to the

injection site, but also exhibited accumulation in liver, spleen, kidney and, to a lesser extent, adrenal gland, duodenum, heart, and lung. Accumulation of si3033 in the remaining tissues was low or undetectable (Fig. 3b and Supplementary Fig. 2b). To confirm the accumulation of si3033 in injection site skin, liver, spleen, and kidney, we visualized si3033 using an in situ hybridization miRNAscope assay (Supplementary Fig. 2c, d). Histopathology examination of these

tissues did not show significant changes (e.g., morphological, architectural, and colorimetric features) in the cells and tissues of si3033 treated mice when compared to PBS control (Supplementary Fig. 2e). To further evaluate the safety of systemic exposure to si3033, we performed blood chemistry and complete blood count (CBC) diagnostics in mice at 24 and 72 h post injection of 20 mg/kg si3033. Blood chemistry and CBC profiles were similar between si3033 and PBS-injected mice except there was a transient decrease of platelet counts in the siRNA-treated mice (Fig. 3c). This finding was likely due to the exposure of a relatively high dose of chemically modified oligonucleotides during the initial phase of clearance; the phenomenon and the underlying mechanism were reviewed previously[34]. Overall, these data suggest that si3033 was well-tolerated and the safety profile is consistent with our previous studies[27,31,35].

### Local administration of si3033 reduces skin infiltration of autoreactive CD8+ T cells and prevents skin depigmentation in a mouse model of vitiligo

To evaluate the therapeutic efficacy of si3033 in vivo, we utilized a mouse model of vitiligo that is based on the adoptive transfer of autoreactive CD8+ T cells that target melanocyte-specific premelanosome protein (PMEL) of recipient mice[36,37]. To induce disease, irradiated recipient mice were co-injected with PMEL CD8+ T cells and PMEL cognate antigen-pulsed bone marrow-derived dendritic cells[38,39]. This model mimics the effector phase of skin depigmentation and recapitulates human vitiligo progression, including activated IFN-γ signaling, upregulation of MHC class I molecules on melanocytes, and increased skin infiltration of autoreactive CD8+ PMEL T cells[29]. Antibodies targeting IFN-γ signaling molecules, CXCL10, CXCR3, and IFN-γ, as well as JAK inhibitors, are effective treatments in this model[39–42]. Upon vitiligo induction, mice rapidly develop epidermal depigmentation within 3–7 weeks in four major body sites: footpads, nose, ears, and tail.

Due to the symmetrical nature of epidermal depigmentation in this vitiligo model, we were able to inject si3033 into the left footpad and siNTC into the right footpad of each mouse to avoid interindividual variability and ensure consistent therapeutic readout. As a first step to optimize the dosing regimen, we injected the maximal delivery dose (0.08 mg in 30 μL PBS; at ~200 μM) of fluorescently labeled si3033 (left footpad) into wild-type mice and confirmed that injection into one footpad did not cause cross-delivery to the other footpad (Supplementary Fig. 3a, b). The silencing level of JAK1 mRNA in footpad skin was then measured over 5 weeks. We observed up to 25% silencing of JAK1 for a duration of two weeks compared to siNTC control (Fig. 3d). To determine whether this level of JAK1 silencing is sufficient to inhibit IFN-γ signaling, we re-injected si3033 or siNTC into a separate cohort of wild-type mice, collected 3-mm skin punches from footpads 2 weeks post injection, and incubated the skin punches with recombinant mouse IFN-γ. si3033 significantly reduced expression of IFN-γ-inducible chemokines, CXCL9 and CXCL10 (Fig. 3e). CXCL11 was not measured because it is not produced in wild-type C57BL/6 J mice due to a frameshift mutation of the *Cxcl11* gene[43–45]. Collective results from these wild-type mouse experiments indicate a repeat injection scheme (every 2 weeks) might be required to maintain functional JAK1 silencing in the footpad skin. For experiments in the vitiligo model, we therefore prophylactically administered to mice with two doses (0.08 mg/dose) of si3033 and siNTC. The first dose was given 2 weeks prior to vitiligo induction and the second dose was given just prior to vitiligo induction on day 0 (Fig. 3f). Skin infiltration of PMEL CD8+ T cells and skin depigmentation were both significantly decreased in the si3033-treated footpad compared to the siNTC control (Fig. 3g, h and Supplementary Fig. 4). The distribution profiles of pigment intensity of footpad images were analyzed and visualized (Supplementary Fig. 5). These data indicate 20–30% silencing of JAK1 in the skin could be sufficient

to inhibit autoreactive CD8+ T-cell migration to the skin and prevent autoimmunity.

We were surprised to find that the targeting site of si3033 is highly conserved between human and rat JAK1 mRNA transcripts with only a single-nucleotide mismatch. The seed region (i.e., 2nd–7th or 8th nucleotide of antisense strand) of the siRNA is fully complementary to rat JAK1 mRNA, indicating si3033 might also be active for silencing rat JAK1 (Supplementary Fig. 6a). To test this hypothesis, we intradermally injected rat skin with 0.13 mg/dose of si3033 and siNTC and quantified the JAK1 mRNA expression 1-week post injection. si3033 significantly silenced rat JAK1 mRNA compared to siNTC, for approximately 60% (Supplementary Fig. 6b, c). This pilot study creates opportunities to evaluate the modulation of JAK1 in applicable rat models of autoimmune and inflammatory skin diseases. A full or partial alignment of the targeting site sequence (i.e., UGAAGUACAAAGGAAUCUGC) of si3033 to available sequencing data of JAK1 transcripts might expand the targetability of si3033 in additional species for functional genomic and therapeutic studies; however, the activities of si3033 in these species necessitate individual validation in relevant experimental conditions.

### si3033 accumulates in major cell types and inhibits JAK1-dependent IFN-γ signaling in human skin

Intradermal injection of DCA-siRNAs into human skin (a viable route of administration for treating skin diseases) should, in principle, further enhance the local retention of DCA-siRNAs and incur minimal systemic effects. To test intradermal delivery of si3033, we injected 50 μL of fluorescently labeled si3033 into human skin explants. We found that si3033 exhibited a lateral spreading diameter of ~8-mm after 24 h (Fig. 4a), broadly distributing to both epidermis and dermis (Fig. 4b). Because cellular uptake of chemically modified siRNAs varies by tissue and cell type, we determined the relative uptake efficiency of si3033 in major human skin cell types using flow cytometry. We found that si3033 accumulated to the highest degree in fibroblasts, followed by melanocytes, antigen-presenting cells (APCs), endothelial cells, NK cells, granulocytes, T cells, and to the least degree, in keratinocytes (Fig. 4c and Supplementary Fig. 7). This observation might be attributed to varying rates of endocytosis and siRNA trafficking across different cell types[46].

The silencing of JAK1 in si3033 injected (0.13 mg) skin biopsies was measured after 4 days of culture ex vivo. si3033 resulted in ~62% silencing of JAK1 mRNA (Fig. 4d). When we quantified JAK1 silencing in enzymatically isolated epidermis and dermis, we found that si3033 led to ~47% silencing of JAK1 in the epidermis and ~69% silencing in the dermis (Fig. 4e). This difference in JAK1 silencing level between the two skin layers is likely due to the depth of injection, differing JAK1 expression levels, and si3033 uptake in each cell type. Other factors also potentially impact the silencing efficacy, such as subcellular localization of target mRNAs (which impacts accessibility to RISC) in different cell types[47].

We next questioned how the level of silencing (~62%) of JAK1 in human skin may prevent activation of JAK1-dependent inflammatory signaling. Four days after si3033 injection, we stimulated skin biopsies with recombinant human IFN-γ, then measured mRNA expression levels of JAK1-dependent IFN-γ-inducible chemokines (CXCL9, CXCL10, and CXCL11) after 24 h. si3033 enabled significant downregulation of all three chemokine mRNAs (Fig. 4f), with corresponding decreases in expression at the protein level (Fig. 4g). These data indicate that JAK1 silencing by si3033 may translate to functional inhibition of IFN-γ related inflammatory responses in human skin.

### si3033 supports broad modulation of IFN-γ mediated inflammatory responses in human skin ex vivo

Binding of IFN-γ to its receptor activates the JAK1/2-STAT1 pathway, eliciting diverse biological effects through interacting with

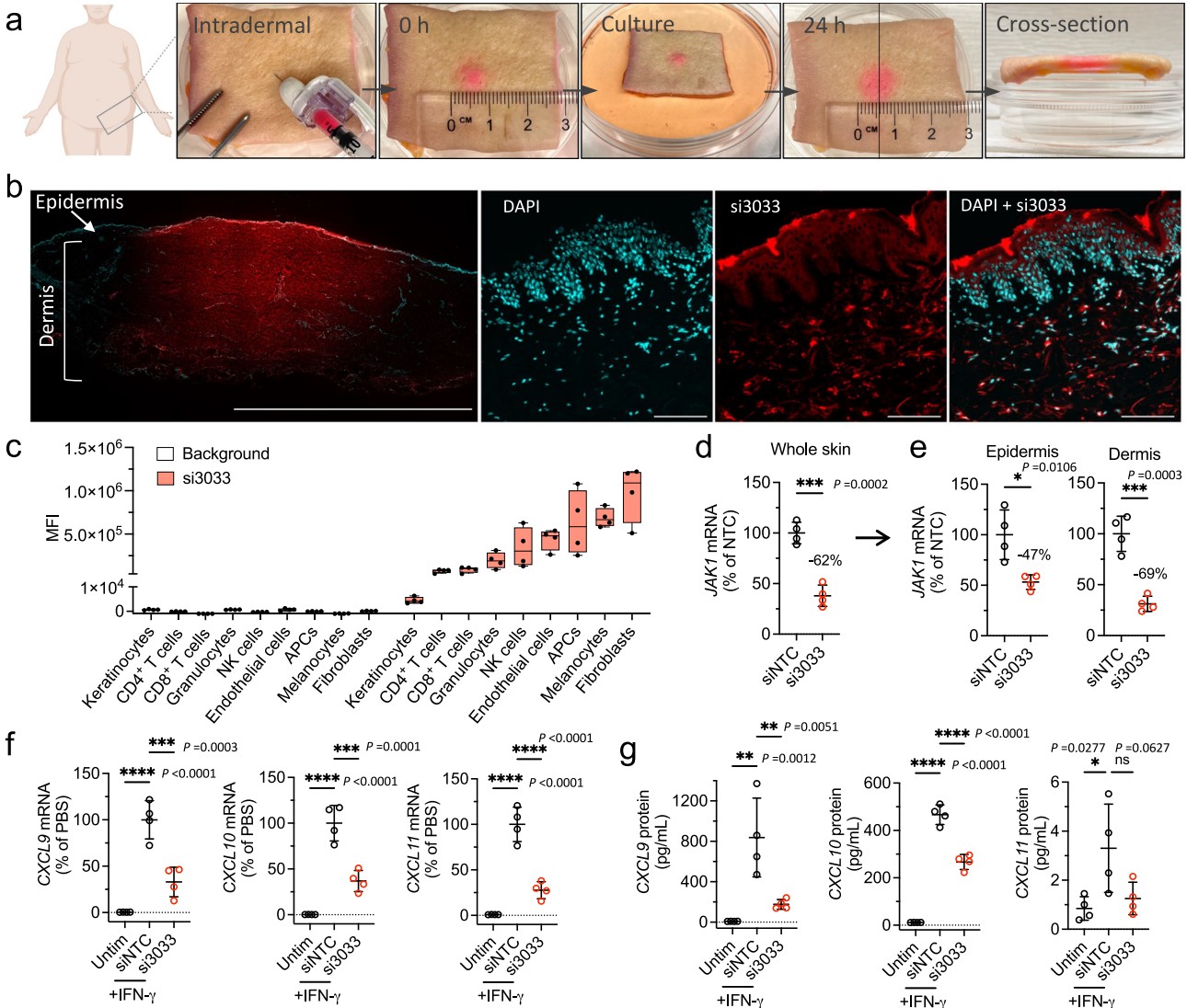

**Fig. 4 | si3033 efficiently downregulates JAK1-dependent inflammatory signaling in human skin ex vivo. a** Intradermal injection of 200 μM (0.13 mg in 50 μL of PBS) fluorescently (Cy3)-labeled si3033 into human skin explant. **b** Fluorescent microscope images of cross-sectioned skin. Skin was cultured for 24 h before microscopy. Nuclei were stained with DAPI in cyan, and si3033 was in red. Left to right: image 1: scale bar length 10 mm; image 2 to 4: scale bar length 100 μm. **c** Relative uptake efficiency of si3033 in skin cell types of human skin (*n* = 4 biologically independent samples; Min to max values; box edges represent quartiles and mean). 0.013 mg of Cy3-si3033 in 50 μL of PBS (20 μM) was intradermally injected to reduce the excess fluorescence intensity of siRNA for flow cytometry analysis. **d** JAK1 silencing in whole skin and **e** in epidermis and dermis separately.

Skin biopsies (8 mm) were injected with 0.13 mg of si3033 for 4 days; epidermis and dermis were separated by 1 h of dispase digestion. JAK1 mRNA level was quantified by QuantiGene 2.0 assay (*n* = 4 biologically independent samples, mean ± s.d.; two-sided unpaired *t* test, \**P* < 0.05, \*\*\**P* < 0.001). **f** mRNA and **g** protein levels of IFN-γ-inducible chemokines CXCL9, CXCL10, and CXCL11. Skin biopsies were stimulated with 10 ng/mL of recombinant IFN-γ and 10 ng/mL of TNF-α (for synergy) for 24 h, mRNAs were quantified by QuantiGene 2.0 assay and proteins were measured by enzyme-linked immunosorbent assay (*n* = 4 biologically independent samples, mean ± s.d.; one-way ANOVA, \**P* < 0.05, \*\**P* < 0.01, \*\*\**P* < 0.001, \*\*\*\**P* < 0.0001, ns not significant). Source data are provided as a Source Data file.

intracellular signaling networks and regulating the expression of a large number of genes beyond CXCL chemokines[48]. For example, IFN-γ-JAK1/2-STAT1 signaling induces upregulation of interferon regulatory factors (IRFs), a class of transcription factors that activate interferon-simulated response elements and leads to the transcription of secondary response genes[49–51]. To study how JAK1 silencing by si3033 modulates the broader effects of IFN-γ signaling activation in human skin, we intradermally injected PBS, siNTC, or si3033 in skin biopsies ex vivo, then stimulated the samples with IFN-γ and performed RNA sequencing. We found that IFN-γ stimulated skin exhibited broad inflammatory responses, including strong induction of IFN-γ-inducible signature genes CXCL9, CXCL10, and CXCL11 as well as interferon-induced guanylate-binding proteins GBP1, GBP4, and GBP-5 (Fig. 5a). Gene set enrichment analysis showed a strong upregulation

of gene sets related to innate immunity and interferon response upon stimulation (Fig. 5b).

We next performed a likelihood ratio test for an interaction between IFN-γ stimulation and si3033 treatment, followed by k-means clustering, to assess how JAK1 silencing alters the response to IFN-γ stimulation. Clustering genes for which the interaction between stimulation and JAK1 silencing was significant (FDR ≤ 0.05) resulted in one cluster containing genes whose downregulation upon IFN-γ stimulation was minimized in si3033-treated samples (Cluster 1), and two clusters containing genes whose upregulation upon IFN-γ stimulation was minimized in the si3033-treated samples (Clusters 2 and 3) (Fig. 5c and Supplementary Fig. 8). Gene ontology analyses revealed Cluster 1 was enriched for genes related to extracellular matrix organization and angiogenesis, whereas Clusters 2 and 3 were enriched for genes related

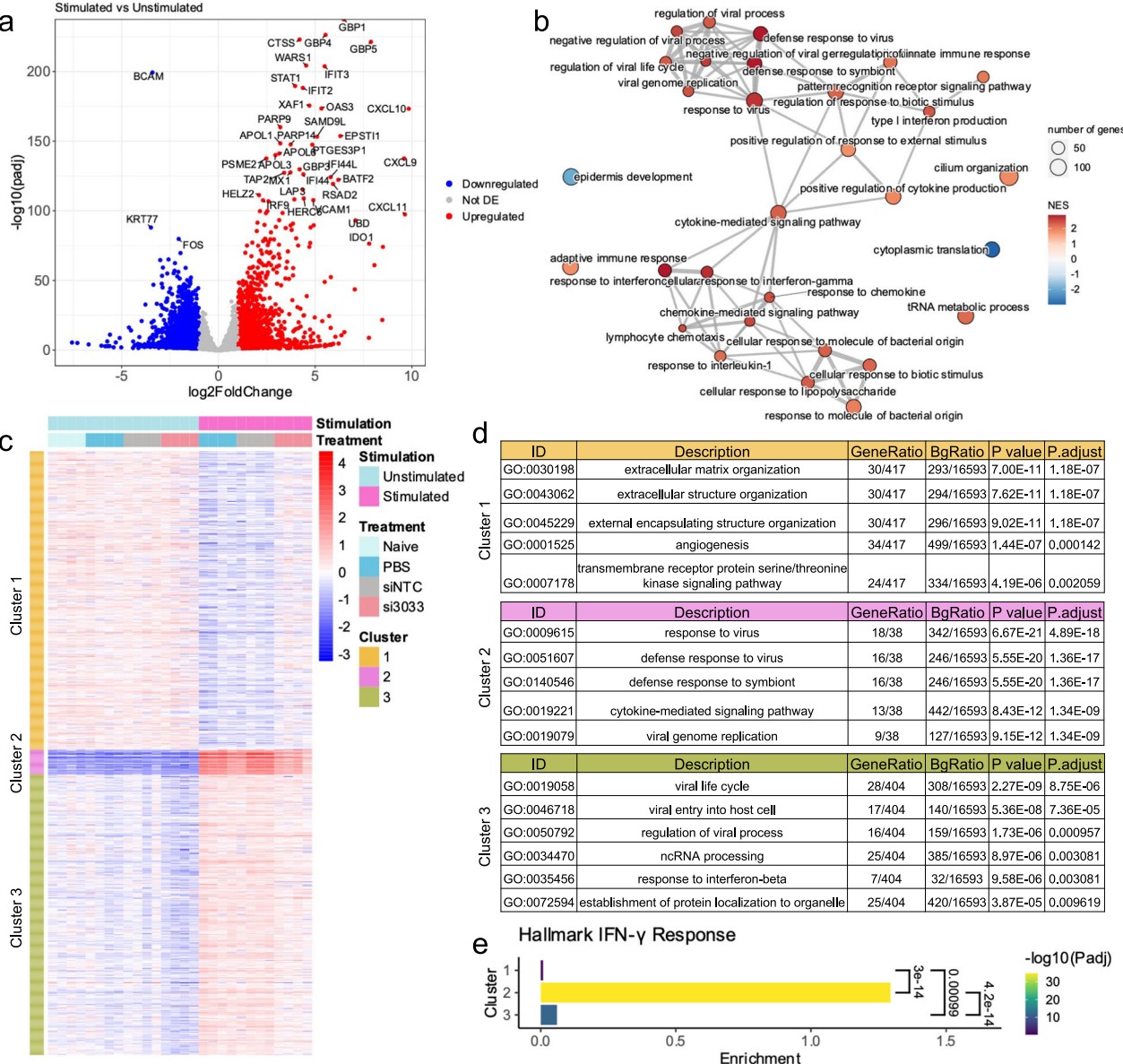

**Fig. 5 | si3033 alleviates IFN-γ-induced inflammatory responses in human skin explant.** Human skin biopsies were injected with PBS, 0.13 mg siNTC or si3033 and cultured for 4 days ex vivo (unstimulated; naive: non-injected control); IFN-γ signaling in a separate set of samples was induced with 10 ng/mL of recombinant IFN-γ and 10 ng/mL of TNF-α for 24 h (stimulated). **a** Volcano plot showing differentially expressed genes with IFN-γ stimulation. PBS and siNTC-treated samples were analyzed together, controlling for treatment status. (Genes colored: blue if FDR ≤ 0.05 and log2FoldChange ≤ −1, red if FDR ≤ 0.05 and log2FoldChange ≥ 1, gray if either FDR > 0.05 or −1<log2FoldChange < 1, n = 4 biologically independent samples per treatment). padj: Benjamini−Hochberg-adjusted P value as calculated by a two-sided Wald Test implemented within DESeq2. **b** Gene set enrichment analysis for the differential expression analysis in (**a**). Gene ontology terms were clustered by semantic similarity and nodes were colored by the normalized enrichment score. **c** Heatmap showing k-means cluster results of genes with a significant

interaction between IFN-γ signaling stimulation and si3033 treatment (Likelihood ratio test, FDR ≤ 0.05, n = 4 biologically independent samples per treatment). **d** Most significant terms from gene ontology analysis of genes in each cluster from k-means clustering in (**c**). **e** Enrichment analysis of genes in the Hallmark IFN-γ Response in each cluster from k-means clustering in (**c**) with Enrichment defined as the number of genes in a cluster belonging to Hallmark IFN-γ Response divided by the number of genes in that cluster not belonging to Hallmark IFN-γ Response. Statistical analysis for the enrichment of Hallmark IFN-γ Response genes in each cluster was conducted using a hypergeometric test in R (phyper, lower.tail=FALSE) with Benjamini−Hochberg correction for multiple comparisons. Statistical analysis to compare enrichment between clusters was performed using a binomial family generalized linear model with a logit link, and multiple test-corrected P values were calculated by a two-sided Tukey HSD test using the emmeans R package.

to innate immune response and cytokine signaling (Fig. 5d). Cluster 2, specifically, was enriched for genes in Hallmark IFN-γ Response ontology compared to Cluster 3 (Fig. 5e), indicating that Cluster 2 reflects the direct response to IFN-γ signaling stimulation. Analysis of select genes downstream of IFN-γ signaling consistently showed a reduced response to IFN-γ stimulation in si3033-treated samples (Supplementary Fig. 9). Interestingly, expression of JAK1 was stable in stimulated versus unstimulated samples, whereas JAK2 and STAT1

were noticeably upregulated upon IFN-γ stimulation, indicating silencing JAK1 might provide a more consistent modulation of downstream effects of the IFN-γ pathway.

Taken together, the transcriptome analyses demonstrate that si3033 effectively alleviates the transcriptional response of human skin to IFN-γ signaling activation. These observations could inform therapeutic intervention of other JAK1-dependent inflammatory responses in human skin.

## Discussion

Here, we report the rational design of a JAK1-selective siRNA, providing a framework for the development of targeted therapies using RNAi technologies. Our developed siRNA potently silences human, non-human primate, mouse, and rat JAK1 mRNA expression, demonstrates therapeutic efficacy in a mouse model of vitiligo, and reduces the IFN-γ-dependent inflammatory response in human skin ex vivo. The JAK1 siRNA could be used to treat a large variety of skin diseases driven by JAK1-mediated inflammatory signaling. Moreover, further chemical engineering and pharmaceutical formulation of the JAK1 siRNA could expand its scope of use for the treatment of immune disorders beyond the skin.

Targeted treatment of vitiligo currently relies on the small molecule JAK1/2 inhibitor ruxolitinib (Opzelura) or off-label use of tofacitinib (a small molecule JAK1/3 inhibitor). Both treatments are effective[52,53]—indicating JAK1 is a key therapeutic target for vitiligo—but currently have black box warnings on their labels due to safety concerns associated with inhibiting multiple JAKs. We demonstrate that, by harnessing a natural mechanism of gene silencing (RNAi), synthetic siRNA therapeutics can selectively silence JAK1 to inhibit IFN-γ-JAK1/2-STAT1-CXCL9/10/11 signaling and mitigate vitiligo pathogenesis. Our findings indicate that selective JAK inhibitors with a narrow spectrum of action on cytokines may represent a viable and safer path toward the treatment of certain autoimmune and inflammatory disorders. This strategy may be particularly important for systemic JAK inhibition when treating multisystem immune disorders.

A critical step toward clinical implementation of siRNA therapeutics for the treatment of skin diseases is to assess the safety and pharmacokinetic properties of the compounds in animal models that closely resemble human skin characteristics. We have found, in this and previous work, that a fraction of injected DCA-siRNAs distribute systemically into clearance organs (i.e., liver, spleen, and kidney)[27,31], which raises concerns about unintended systemic effects. Mouse skin, by nature, is thinner and more permeable than human skin. Therefore, it is reasonable to predict better local retention (and therefore less systemic exposure) of intradermally injected DCA-siRNAs in human skin versus mouse skin. If so, the duration of effect of DCA-siRNAs in human skin is expected to be longer than what we observed in mouse skin. In addition, biologically compatible chemical modifications of 3' phosphate backbone and 5' phosphate of antisense strand should provide extra metabolic stability of siRNA; thus allowing for prolonged efficacy and less frequent dosing in the clinic. These hypotheses need be tested in future preclinical studies using relevant animal models for skin research, such as pigs. Such studies could be carried out using one of the lead siRNAs identified in this study (i.e., 883), which is cross-targeting to pig JAK1. Future work in pig skin models would provide invaluable insights into the therapeutic potential of the JAK1 siRNA.

The vitiligo mouse model in this study mimics many molecular features of human vitiligo, including upregulation of signature IFN-γ pathway genes (i.e., IFN-γ, CXCL9, and CXCL10). However, this model has an early onset of autoimmunity after vitiligo induction that leads to rapid epidermal depigmentation (within 3–7 weeks), whereas repigmentation is slow. It was challenging to test the therapeutic efficacy of si3033 administered after disease manifestation in this model because the accelerated autoimmunity in the early stage of vitiligo progression overpowers the delayed effects of si3033 (maximum JAK1 silencing required 1–2 weeks), thereby interfering with final therapeutic read-outs. Future work using a skin disease model with milder autoimmunity that more closely resembles human disease progression may be helpful for evaluating the disease-reversing effect of the JAK1-siRNA. Moreover, dosing regimens of the JAK1 siRNA in each specific disease might necessitate a unique optimization process.

We demonstrate that 20 mg/kg of si3033 is well-tolerated in mice in vivo. Our data in the human skin explant revealed that intradermal injection of 0.13 mg si3033 (in 50 μL PBS) enables efficient JAK1 silencing in a skin area of ~8-mm diameter. The systemic exposure of si3033 associated with this dosing level, even with multiple injections, might have a wide therapeutic index for patients in the normal range of body weight. Our approach may therefore be particularly suitable for intralesional modulation of JAK1 for certain skin diseases, such as patchy hair loss in alopecia areata. The prolonged duration of effect of si3033 may enable less frequent dosing of patients to further improve safety profiles, a few clinical visits yearly might be sufficient to maintain therapeutic efficacy.

Topical use of small molecule ruxolitinib is patient friendly, however, requires twice daily application to the affected skin areas of only up to 10% body surface due to safety concerns, satisfactory patient response may require treatment for more than 24 weeks. siRNAs differ from small molecules in action of mechanism. Upon cellular internalization, entrapment of metabolically stabilized siRNAs in lysosomal and endosomal compartments generates an intracellular depot of the drug with slow release, which offers improved potency and duration of effect. For skin conditions that involve large area of body surface that intradermal injection of siRNA is not straightforward, depth-controlled transdermal delivery methods would be beneficial and is an area being actively explored (e.g., microparticle and skin-penetrating formulations)[54,55].

Our RNAseq data highlight the potential of si3033 to modulate a broad range of inflammatory responses upon IFN-γ stimulation in human skin; additional work will be required to dissect the complexity of these biological events. A clearer understanding of these events, especially under pathological conditions in a cell-type-specific resolution could better inform the therapeutic intervention of many diseases driven by dysregulated IFN-γ signaling. Correspondingly, more efficient delivery of siRNA drugs into skin cell types that drive disease progression would be advantageous, this may be achieved through antibody conjugates against cell-type-specific markers or small RNA trafficking mechanisms.

In summary, we show the adaptability of RNAi-based design of targeted therapies and the promise of functional delivery via bioconjugates to the skin, indicating that siRNAs could become an ideal therapeutic paradigm for gene silencing to treat skin diseases. Importantly, as JAK1 mediates a variety of inflammatory cytokine signaling pathways, the validated JAK1 siRNAs presented in this work could be further engineered for immunomodulation of additional pathological processes in other tissues, offering unprecedented therapeutic opportunities. The pipeline presented in this work lays a foundation for future development of disease-modifying RNA drug modalities.

## Methods

### Design of JAK1 siRNAs

siRNAs were designed based on a 20-nucleotide target sequence from human (accession number: NM_001321852) and mouse (NM_146145) JAK1 transcripts. siRNA sequences were excluded if they had: (1) > 56% of GC content; (2) single-nucleotide stretches of four or more; or (3) perfect homology to human miRNA seeds at position 2–7 of the antisense strand[56]. To minimize transcriptomic off-target effects, siRNAs were excluded if position 2–17 of the antisense strand had full complementarity to non-target mRNAs (except other JAK1 isoforms). Targeting sequences (along with +10/−15 flanking nucleotides) were scored using a weight matrix[57] and top-scoring sequences in each species were identified. Cross-species targeting was determined based on perfect homology of the 16-nucleotide homology region within the target sequence (positions 2–17) to the target sequence of the other species. siRNAs were named based on the position in the target transcript from which they were extracted (e.g., human siRNA 3033 was extracted from positions 3033–3052 of NM_001321852); for cross-species targeting sequences, naming was based on the human transcript.

## RNA oligonucleotide synthesis

Compounds for in vitro screening were synthesized using standard solid-phase phosphoramidite chemistry on a Dr. Oligo 48 high-throughput RNA synthesizer (Biolytic); compounds for in vivo injections were made using a MerMade 12 (BioAutomation) synthesizer. Standard RNA 2′-O-methyl, 2′-fluoro modifications were applied to improve siRNA stability (Chemgenes). The sense strands of in vitro compounds were synthesized in a 1-μmol scale on a cholesterol-conjugated solid support (Chemgenes), and the sense strands of in vivo compounds were synthesized at 5-μmol scale on an in-house synthesized docosanoic acid-functionalized controlled pore glass (CPG)[31,35,58]. Antisense strands were synthesized on CPG functionalized with a Unylinker (Chemgenes), bis-cyanoethyl-N, N-diisopropyl CED phosphoramidite (Chemgenes) was used to introduce a 5′-mono-phosphate for in vitro experiments, and custom 5′-(E)-vinylpho-sphonate phosphoramidite (Chemgenes) was applied for in vivo studies. Cy3-phosphoramidites (Gene Pharma) were used for fluorescence labeling of the 5′ of sense strands.

## Oligonucleotide deprotection

For post-synthesis deprotection, sense strands were cleaved from the CPG and deprotected using 40% aq. methylamine and 30% $NH_4OH$ (1:1, v/v) at room temperature for 2 h. Antisense strands were cleaved and deprotected with 30% NH4OH containing 3% diethylamine for 20 h at 35 °C. The deprotected oligonucleotide solutions were filtered to remove CPG residues. The filtrates were immediately frozen in liquid nitrogen and dried by a SpeedVac vacuum centrifuge in chemical hood. The resulting pellets were reconstituted in 5% acetonitrile for subsequent purifications.

## HPLC purification

Oligonucleotide purification was carried out on an Agilent 1290 Infinity II system. Sense strands were purified using a reverse phase preparative column (Hamilton PRP-C18) under the following conditions: buffer A, 50 mM sodium acetate in 5% acetonitrile; buffer B, 100% acetonitrile; the gradient was 0–20% for 3 min, 20–70% for 23 min, then cleaned and recalibrated for 9 min; column temperature was 60 °C and the flow rate 40 mL/min. Antisense strands were purified using an anion-exchange column (with SOURCE™ 15Q) under the following conditions: buffer A, 10 mM sodium acetate in 20% acetonitrile; buffer B, 1 M sodium perchlorate in 20% acetonitrile; the gradient was 0–20% for 3 min, 20–70% for 23 min, then cleaned and recalibrated for 9 min; column temperature was at 55 °C; and the flow rate was 40 mL/min. The oligonucleotides were detected by measuring peaks with UV absorbance at 260 nm. Peak fractions were automatically collected for confirming the identities. The oligonucleotide fractions were quality-controlled by liquid chromatography-mass spectrometry (LC/MS), and pure fractions were combined, frozen, and dried in a Speed Vacuum centrifuge overnight. The dried oligonucleotides were resuspended into water and desalted by size-exclusion chromatography (with Sephadex G-25 Fine).

## LC-MS analysis

The purity and identity of all oligonucleotides were analyzed on an Agilent 6530 accurate mass Q-TOF LC-MS system using ion-pair reverse phase chromatography (LC column: Agilent $2.1 \times 50$ mm AdvanceBio C18 oligonucleotide) under the following conditions: buffer A: 9 mM triethylamine/100 mM hexafluoroisopropanol in water; buffer B: 9 mM triethylamine/100 mM hexafluoroisopropanol in methanol; the column temperature was 60 °C and flow rate 0.5 mL/min; Peaks were detected by measuring UV at 260 nm. MS parameters were: ion source, electrospray ionization; ion polarity, negative mode; mass scan range, 100–3200 $m/z$; scan rate, 2 spectra per second; capillary voltage, 4000 V; fragmentor voltage, 180 V.

## Cell culture

HeLa cells (ATCC; #CCL-2) were maintained in Dulbecco's Modified Eagle's Media (Corning Cellgro; #10-013CV). Neuro-2a (N2a) cells (ATCC; #CLL-131) were maintained in Eagle's Minimum Essential Media (ATCC; #30-2003). HEL 92.1.7 cells (ATCC; TIB-180) were maintained in RPMI-1640 media (ATCC; #30-2001). All media were supplemented with 10% fetal bovine serum (Gibco; #26140) to make complete growth media, and all cells were grown at 37 °C with 5% of $CO_2$ supply. No antibiotics were used in the studies. Cells were split every 2–4 days and discarded after 15 passages.

## In vitro screening

HeLa and N2a cells were treated with cholesterol-conjugated siRNAs at 1.5 μM, which also served as the maximal dose for dose–response assays, for 72 h in 3% FBS media that was made from 50/50 volume of 6% FBS media/Opti-MEM media (Gibco, #31985–079). Cells were harvested in diluted lysis mixture consisting of a 1:2 ratio of lysis mixture (Invitrogen, #13228) to water and 0.2 mg/mL of proteinase K (Invitrogen, #25530–049), then lysed at 55 °C for 30 min. The target mRNA levels were measured by using Quantigene 2.0 assays (Affymetrix). All QuantiGene detection probesets were ordered from ThermoFisher: human *JAK1* (#SA-50455), mouse *Jak1* (#SB-3029714). JAK1 mRNA expressions were normalized to housekeeping genes and the probesets used were human *PPIB* (#SA-10003), human *HPRT* (#SA-10030) or mouse *Ppib* (#SA-10002), and mouse *Hprt* (#SA-15463).

## Wes ProteinSimple assay

HeLa and N2a cells were harvested and lysed in Pierce RIPA buffer (ThermoFisher; #89901) with cOmplete proteinase inhibitor cocktail (Roche; #11697498001). Total protein levels were determined using a Bradford assay (ThermoFisher; #23236); Samples were diluted in 0.1× sample buffer (Bio-techne) and a total of 1.2 μg protein per sample was loaded for antibody staining. The primary antibody for human and mouse JAK1 was monoclonal anti-JAK1 (Cell Signaling; #50996) in a 1:500 dilution; and for human and mouse housekeeping β-actin was monoclonal anti-β-actin (Cell Signaling; #4970) in a 1:500 dilution. Secondary antibodies used were: Anti-mouse detection module (Bio-techne; #DM-002) and anti-rabbit detection module (Bio-techne; #DM-001). The assay was performed as described by the ProteinSimple protocol using the 16–230 kDa separation module (Bio-techne; SM-W004) on a Wes system (Bio-techne). For in vivo studies, footpad skin biopsies were mechanically homogenized in 500 μL Pierce RIPA buffer proteinase inhibitor cocktail, the processed skin homogenate was centrifuged at 10,000×*g* for 5 min, and the clear supernatant was collected for Wes ProteinSimple assay. The primary antibody for mouse JAK1 was monoclonal anti-JAK1 (Cell Signaling; #50996) in a 1:500 dilution; and for mouse housekeeping β-actin was monoclonal anti-β-actin (Cell Signaling; #4970) in a 1:250 dilution.

## JAK subtype selectivity study

Human HeLa cells were treated with cholesterol-conjugated JAK1 lead compound 3033 at 1.5 μM at 37 °C for 72 h. The expression of human JAK1, JAK2, JAK3, and TYK2 were measured by the Quantigene 2.0 assay with the following probesets (ThermoFisher): human *JAK1* (#SA-50455) human *JAK2* (#SA-10038), human *JAK3* (#SA-10158), human *TYK2* (#SA-3003840), and human *ACTB* (#SA-10008) as a housekeeping gene. Small molecule JAK1&2 inhibitor ruxolitinib was purchased from TOCRIS (#7064, MW: 310.87 Da, $C_{17}H_{18}N_6\cdot1/4 \ H_2O$). Ruxolitinib was reconstituted according to the manufacturer's instructions. To compare the potency of JAK1 siRNA 3033 over ruxolitinib in inhibiting the stimulation of IFN-γ signaling, HeLa cells were treated with siRNA or ruxolitinib for 72 h at a concentration of 1.5 μM followed by replacing the media containing 10 ng/mL recombinant human IFN-γ protein (R&D System, #285-IF-100) and 10 ng/mL recombinant human tumor necrosis factor (TNF)-α protein (R&D System, #210-TA-020). The

expression of mRNA was measured by the Quantigene 2.0 assay with the following probesets (ThermoFisher): human *JAK1* (#SA-50455) human *CXCL9* (#SA-12372), human *CXCL10* (#SA-50393), human *CXCL11* (#SA-50464), and human *ACTB* (#SA-10008) as a housekeeping gene.

## Luciferase reporter

GAS reporter (Luc)-HeLa cell line (IFN-γ/JAK/STAT1 pathway) was purchased from BPS Bioscience (#79041). The cell line was maintained in Eagle's Minimum Essential Media (ATCC; #30-2003) supplemented with 10% FBS, 100 U/ml penicillin–streptomycin, and 800 μg/mL of geneticin (Life Technologies, #11811031). si3033 was transfected into GAS reporter cells by using Lipofectamine™ RNAiMAX Transfection Reagent (Invitrogen, #13778150). ONE-STEP luciferase assay system (BPS Bioscience, #60690-2) was used to detect firefly luciferase activity according to the manufacturer's protocol. The Luciferase signal was measured on a SpectraMax M5 microplate reader. The expression of JAK1 and JAK2 mRNA was quantified by Quantigene 2.0 assay with the following probesets: human *JAK1* (#SA-50455), human *JAK2* (#SA-10038), and human *ACTB* (#SA-10008) as housekeeping control.

## Animals

Mouse studies were performed in accordance with the guidelines of the University of Massachusetts Chan Medical School Institutional Animal Care and Use Committee (IACUC). All procedures were approved under the Protocol #202000010 (Khvorova Laboratory) and #201900330 (Harris Laboratory) and in accordance with the National Institutes of Health Guide for the Care and Use of Laboratory Animals. T-cell donor strain B6.Cg-Thy1a/Cy Tg(TcraTcrb)8Rest/J, SCF strain B6.Cg-Tg(KRT14-Kitl*)4XTG2Bjl/J, wild-type strain C57BL/6J of mice (both male and female) were purchased from The Jackson Laboratory (Bar Harbor, ME). Mice were 8–12 weeks of age at the time of the experiments. The female Lewis rats were purchased from Charles River Laboratory (Wilmington, MA) and were at 8–10 weeks of age at the time of the experiment. The colonies were maintained and housed at pathogen-free animal facilities at UMass Chan Medical School with 12 h light/12 h dark cycle at controlled temperature (23 ± 1 °C) and humidity (50% ± 20%) with free access to food and water.

## Peptide nucleic acid (PNA) hybridization assay

Tissue biodistribution levels of siRNA were measured using a PNA hybridization assay[31,59]. The accumulation of si3033 was quantified using a custom Cy3-labeled fluorescent PNA oligonucleotide probe (Cy3-OO-GTACAAAGGAATCTGA-KK, PNABio) that is fully complementary to the antisense strand. Pre-weighed tissue punches were placed in 500 μL homogenizing solution (Invitrogen; QS0517) containing 0.2 mg/mL proteinase K (Invitrogen, #AM3546) in a QIAGEN collection microtube holding a 3-mm tungsten bead. The tissues were then homogenized for 10 min under 30 Hz of frequency using a QIA-GEN TissueLyser II. Homogenized tissues were incubated at 55 °C for 30 min and centrifuged at 1000×g for 10 min. Sodium dodecyl sulphate was precipitated from homogenates by adding 20 μL of 3 M potassium chloride followed by centrifugation for 15 min at 5000×g. siRNAs in the clear supernatant were hybridized to the PNA probe under 95 °C followed by slow cool down. Anion-exchanged chromatography was used to analyze the sample mixtures on an Agilent 1260 Infinity quad-pump HPLC with a 1260 FLD fluorescent detector. DNA-Pac PA100 column (ThermoFisher) was used for the peak separation. The mobile phases consist of buffer A: 50% water, 50% acetonitrile, 25 mM Tris-HCl (pH 8.5), and 1 mM ethylenediaminetetraacetate; and buffer B: 800 mM sodium perchlorate in buffer A. Gradient conditions: 10% buffer B for 4 min, 50% buffer B for 1 min, then 50% to 100% buffer B for 5 min. Cy3 fluorescence was monitored and peaks were integrated. Final concentrations of oligos were determined using a calibration curve generated by spiking known quantities of si3033 into tissue lysates from untreated animals.

## Histopathology and miRNAscope

Mice were subcutaneously injected with PBS or 20 mg/kg of si3033 (*n* = 3). Seven days post injection, tissue samples of skin local to the injection site, liver, spleen, and kidney were isolated and fixed in neutral-buffered formalin for 24 h and paraffin-embedded. Cross-sectioning of tissues and H&E staining were performed by the Morphology Research Core at UMass Chan. Histopathology examination of blinded sample groups was independently conducted by a histopathologist in the Department of Pathology at UMass Chan. In situ hybridization (ISH) was performed on 5-μm paraffin sections using a custom probe complementary to si3033 (miRNAscope LS probes, BioTechne) and visualized by miRNAscope LS Reagent Kit Red (Bio-Techne, #324600) on a Leica BOND RX autostainer (Leica Biosystems, Wetzlar, Germany). The stained slides were scanned with a Nanozoomer 2.0 HT (Hamamatsu, Shizuoka, Japan) slide scanner. Accumulation of siRNA was visualized by red chromogen.

## In vivo efficacy

Tissues were collected at the indicated time points and stored in RNA later solution (Sigma-Aldrich; #R0901) at 4 °C overnight. Skin biopsies were mechanically homogenized in 500 μL of homogenizing solution (Invitrogen; QS0517) containing 0.2 mg/mL proteinase K (Invitrogen, #AM3546), followed by incubating at 55 °C for 30 min. The processed skin homogenate was then centrifuged at 14,000×g for 5 min, and the clear supernatant was collected for subsequent assays. The mRNA level of mouse JAK1 expression was quantified using QuantiGene Singleplex assay kit (Invitrogen; #QS0016) with the following probesets: mouse *Jak1* (SB-3029714) and mouse *Actb* (SB-10003).

## Blood diagnostics

C57BL/6J mice were subcutaneously injected with PBS or si3033 at 20 mg/kg. Blood sampling was performed at 24 and 72 h post injection. In total, 200 μL of blood was collected in a lithium heparin-coated BD Microtainer tube (BD, #365965) for blood chemistry test. 100 μL of blood was collected in a $K_2$ EDTA-coated bd Microtainer tube (BD, #365974) for a complete blood count (CBC) test. Blood chemistry and CBC diagnostics were conducted by the Diagnostics Laboratory in the Department of Animal Medicine at UMass Chan Medical School.

## Mouse model of vitiligo

Vitiligo was induced[38] by retro-orbital injection of one million auto-reactive CD8+ T cells and one million of bone marrow-derived dendritic cells (BMDCs) into sublethally irradiated (500 rads 1 day before transfer) B6.Cg-Tg(KRT14-Kitl*)4XTG2Bjl/J hosts (The Jackson Laboratory; #009687). Autoreactive CD8+ T cells (PMELs) were purified from the spleen of PMEL TCR transgenic strain B6.Cg-Thy1a/Cy Tg(TcraTcrb)8Rest/J (The Jackson Laboratory; #005023) through negative selection on microbeads (Miltenyi Biotech; #130-095-236) according to the manufacturer's instructions. This transgenic strain carries a rearranged T-cell receptor transgene specific for the mouse homolog (*pmelsi* or *pmel-17*) of human premelanosome protein (referred to as PMEL or gp100), and the T lymphocyte-specific Thy1a (Thy1.1) allele. BMDCs were differentiated following a modified protocol based on the described method[60,61]. Briefly, bone marrow cells were isolated from the femurs and tibias of wild-type mice and were filtered through a 70-μm cell strainer. The red blood cells (RBC) were lysed by RBC lysing buffer (Sigma-Aldrich; #R7757-100ML). The RBC-lysed bone marrow cells were cultured in RPMI-1640 media (ATCC; #30-2001) containing 10% FBS (Gibco; #26140), 100 U/ml of penicillin–streptomycin, 2 mM L-glutamine, 50 μM 2-mercaptoethanol, 20 ng/mL recombinant murine GM-CSF (PeproTech; #315-03), and

10 ng/mL recombinant murine IL-4 (PeproTech; #214-14). Non-adherent BMDCs were harvested and the percentage was determined by flow cytometry. For dendritic cell vaccination, the BMDCs were primed with 10 μM human gp10025–33 (gp100) peptide (GenScript; #RP20344) in Opti-MEM media (Gibco; #11058021) at 37 °C for 3 h and washed with PBS for three times.

## CD8+ PMEL quantification

Skin infiltration of autoreactive PMEL CD8+ T cells in the footpad was quantified by flow cytometry (Cytex Aurora cytometer). Footpad skin was harvested at week 4 after vitiligo induction and treated with RPMI-1640 media (Gibco, #11875093) containing 2.5 mg/mL of collagenase IV (Sigma-Aldrich, #C6885) and 1 mg/mL deoxyribonuclease I (Sigma-Aldrich, #DN25) at 37 °C for 45 min. Skin samples were manually crushed through a 100-μm cell strainer (Fisher Scientific, #22363549). The following dye and flow cytometry antibodies were used for staining CD8+ and Thy1.1+ double-positive PMEL T cells that originated from the donor mice: TruStain FcX PLUS (anti-mouse CD16/32, Biolegend, #156604), Zombie Aqua™ Fixable Viability Kit (Biolegend, #423101), APC-Cyanine7 anti-mouse CD45 (Biolegend, #103116), PerCP-Cyanine5.5 anti-mouse CD8b (Biolegend, #126610), and FITC anti-mouse CD90.1 (Thy1.1, Biolegend, #202503). Cells were fixed by FluoroFix™ fixation buffer (Biolegend, #422101) prior to flow cytometry analysis.

## Human skin explant studies

De-identified surgically discarded fresh abdominal skin was obtained from the UMass Chan Biospecimen, Tissue, and Tumor Bank. The collection of these samples from patient donors was approved by the Institutional Review Board (IRB) at the University of Massachusetts Chan Medical School. All participants gave written informed consent before surgical procedures. The siRNA treatments were conducted within 24 h after the excision of human specimens. Subcutaneous fat was fully removed and all skin specimens were cultured in Iscove's Modified Dulbecco's Media (Sigma-Aldrich; #I3390) containing 10% FBS, 100 U/ml penicillin–streptomycin, and 50 μM 2-mercaptoethanol. In all, 8-mm skin biopsies were taken and injected intradermally with 0.13 mg of siRNAs in a volume of 50 μL. For silencing studies, the skin biopsies were cultured in a 24-well plate with 2 mL of media per punch for 96 h at 37 °C. To induce CXCL9, 10, and 11 expression, after 96 h of siRNA treatment, skin biopsies were treated with new media containing 10 ng/mL recombinant human IFN-γ protein (R&D System, #285-IF-100) and 10 ng/mL recombinant human TNF-α protein (R&D System, #210-TA-020). mRNA expression was measured by the Quantigene 2.0 assay with the following probesets (ThermoFisher): human JAK1 (#SA-50455) human CXCL9 (#SA-12372), human CXCL10 (#SA-50393), human CXCL11 (#SA-50464), and human ACTB (#SA-10008) as a housekeeping gene.

## Dissociation of epidermis and dermis

Human skin biopsies were incubated with 2 mL of 2.5 mg/mL dispase II (Roche, #04942078001) at 37 °C for 1 h. The epidermis and dermis were manually separated. Dermis was further incubated in RPMI-1640 media (Gibco, #11875093) containing 2.5 mg/mL of collagenase IV (Sigma-Aldrich, #C6885) and 1 mg/mL deoxyribonuclease I (Sigma-Aldrich, #DN25) at 37 °C for 45 min. Both the epidermis and dermis were mechanically dissociated.

## Fluorescence imaging

Skin biopsies were embedded into optimal cutting temperature (O.C.T.) compound (Sakura Finetek) and were fresh-frozen before sample sectioning. Tissues were sliced into 4-μm sections that were mounted on glass slides. Nuclei were stained with DAPI. Fluorescent images were acquired with a Leica DMi8 inverted microscope (Leica Microsystems). Images were analyzed using the Leica LAS X software.

## Human skin flow cytometry

8-mm skin punches were injected with 0.013 mg (1/10 of therapeutic dose) of Cy3-labeled si3033 in 50 μL PBS and then cultured for 24 h in 2 mL/punch of Iscove's Modified Dulbecco's Medium (IMDM, Sigma; #I3390-500ML) supplemented with 10% fetal bovine serum (Gibco; #26140), 1× penicillin–streptomycin (Sigma; P4333-100ML), and 50 μM of 2-mercaptoethanol (Sigma; #M3148). Skin punches were digested in IMDM media containing 1.6 mg/mL collagenase, type 4 (Worthington-biochem; #LS004186). Briefly, skin biopsies were diced and minced into small pieces and incubated at 37 °C for 3 h at 200 rpm in gentleMACS C tubes (Miltenyi Biotec, #130-093-237) on a MACS dissociator. The suspended cells were filtered through a 40 μm cell strainer (Fisher Scientific, #22-363-547) and stained with the following dye and fluorescent antibodies: Human TruStain FcX (Biolegend, #422302), Zombie NIR™ Fixable Viability Kit (Biolegend, #423106), PerCP anti-human CD45 (Biolegend, #368506), Brilliant Violet 711™ anti-human CD117 (c-kit) (Biolegend, #313230), Brilliant Violet 785™ anti-human CD90 (Thy1) (Biolegend, #328142), BUV563 anti-Human CD31 (BD Biosciences, #749327), Brilliant Violet 510™ anti-human CD3 (Biolegend, #344828), Brilliant Violet 605™ anti-human CD8 (Biolegend, #344742), Alexa Fluor 700 anti-human CD4 (Biolegend, #317426), BUV737 anti-Human CD56 (BD Biosciences, #612766), Brilliant Violet 570™ anti-human CD16 (Biolegend, #302036). The samples were analyzed using an Aurora cytometer (Cytex Biosciences) and data were processed using FlowJo 10 software.

## ELISA

The protein expression of IFN-γ-inducible chemokines in treated mouse skin biopsies were quantified by using mouse CXCL9/MIG DuoSet ELISA kit (R&D System, #DY492-05) combined with DuoSet Ancillary Reagent Kit 2 (R&D System, #DY008), and mouse CRG-2 (CXCL10) ELISA kit (Invitrogen, #EMCXCL10) according to the manufacturer's instructions. Optical densities were measured using a Tecan M1000 microplate reader, and the absorbance at 540 nm was used to calculate CXCL9 levels using a standard curve derived from known concentrations of the standards. For CXCL10, optical densities at 450 nm were measured on a SpectraMax M5 microplate reader.

## RNA sequencing

Total RNA was extracted from 30 mg of skin sample using RNAeasy Fibrous Tissue Mini Kit (QIAGEN, #74704) following the manufacturer's instructions and the purified total RNA was eluted into RNase-free water for RNAseq. RNA quality control, poly-A enrichment, and library preparation and sequencing were conducted by AZENTA Life Sciences (Illumina platform). Samples were sequenced to a depth of ~30 million paired-end 150 bp reads per sample. The sequenced reads were mapped to the human genome (hg38) using STAR v.2.7.10a under default settings[62] and transcripts were quantified using RSEM v1.3.3 under default settings[63]. Transcript counts were read and summarized to gene level counts using tximport v1.22.0[64] and differential expression analysis was performed using DESeq2 v1.34.0[65] with Bayesian shrinkage applied to log2FoldChange values using apeglm v1.16.0[66] for pairwise comparisons. In pairwise comparisons, genes were considered significantly differentially expressed if padj≤0.05 and |log2FoldChange|≥1 after apeglm shrinkage. For likelihood ratio test analyses, the factors for "Stimulation" and "Injection Status" were combined and a term was created to define whether or not the tissue was treated with si3033. A model matrix was created using the formula "-StimulationandInjection +StimulationandInjection:si3033". The interaction term for UnstimulatedNo_Injection:No_si3033 was then removed from the model matrix, since it defined only the Naïve group, and this model matrix was used as the full model for the likelihood ratio test. The reduced model was the model matrix with all interaction terms removed. Genes were considered significantly differentially expressed if padj≤0.05. Gene set enrichment analysis and gene

ontology analysis were performed, and enrichment map plots were created, using clusterProfiler v4.2.2[67]. K-means clustering was performed on centered, variance stabilizing transformed read counts from genes passing the aforementioned statistical significance cutoff for likelihood ratio test analyses, using the k-means function in R with parameters: centers=3, nstart=50. Heatmaps were created using pheatmap v1.0.12 and all other plots were created using ggplot2 v3.4.0. The.gmt file for HALLMARK_INTERFERON_GAMMA_RESPONSE was downloaded from gsea-msigdb.org and enrichment in clusters was assessed using a hypergeometric test. Comparison of enrichment of HALLMARK_INTERFERON_GAMMA_RESPONSE genes between each of the three clusters was performed using a binomial family generalized linear model with a logit link and pairwise comparisons were performed using emmeans v1.8.4-1.

## Statistical analysis

All data were analyzed using GraphPad Prism 9 software (GraphPad Software, Inc). Detailed sample size and statistical methods used for the data analysis of individual studies can be found in the corresponding figure legends. Differences in all comparisons were considered significant at $P$ values less than 0.05.

## Reporting summary

Further information on research design is available in the Nature Portfolio Reporting Summary linked to this article.

## Data availability

The RNAseq data generated in this study have been deposited to the GEO repository database under accession code GSE234070 [https://www.ncbi.nlm.nih.gov/geo/query/acc.cgi]. All data generated in this study are provided in the Supplementary Information and Source Data file. Source data are provided with this paper.

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

## Acknowledgements

This project was supported by Aldena Therapeutics (sponsored research to J.F.A.), LEO Pharma A/S (sponsored research to A.K. and J.E.H.), and National Institutes of Health (grant no. R35 GM131839 and S10 OD020012 to A.K.; grant no. R01 AR069114 to J.E.H.; K99 AR082987 to Q.T., and grant no. S10 OD028576 to C.E.S.). The project was also supported by Hartford Foundation Vitiligo Grant (to J.E.H.) and the King Trust, Bank of America Private Bank, Co-Trustees Fellowship (to K.A.). We are grateful to Ms. Mary Beth Dziewietin, Ms. Terry Balzano, Ms. Laura Lajoie, and Ms. Andrea Thackeray for their administrative and project management support. We would like to thank Trine Gejsing, Dina Wennike, Gitte Lund, Tinna Chrone Nielsen, and Stinne Ravnsbaek at LEO Pharma for their excellent technical assistance; and the Khvorova, Harris, Rashighi, and Richmond Laboratory members for their continued support, special thanks to Dr. Sarah M. Davis, Dr. Chantal M. Ferguson, and Dr. Ying-Chao Hsueh for their critical advice on the experimental design and data analysis. We would like to thank Dr. Emily Mohn Haberlin for helping revise and proofread this manuscript. Illustration figures were created with BioRender.com under the license of UMass Chan Medical School.

## Author contributions

Q.T., A.K., J.E.H., and A.C. conceived the project. Q.T., J.F.A., C.B.P., and C. Blanchard managed the studies. K.M. performed the computational design of the siRNA sequences for in vitro screening. A.B. developed the docosanoic acid-functionalized oligonucleotide solid support. J.S., H.H.F., Q.T., D.A.C., D.E., B.B., and N.M. synthesized the compounds. Q.T. carried out the in vitro and in vivo studies. S.R.H. analyzed the RNAseq data. K.A. and Q.T. analyzed the flow cytometry data. C. Bartholdy., P.P.S., and M.J. performed the siRNA miRNAscope studies. H.H.F, M.Z.U., K.A., K.G., A.S.M., R.C.F., N.H.S., and M.R. contributed significantly to the ex vivo human skin studies; and V.H., A.S., X.F., K.O., A.S.M., K.G., K.D., S.J.W., J.C., U.Y.A., J.M.R., and S.S. contributed significantly to the in vivo animal studies. Q.T., S.R.H., A.K., and J.E.H wrote the manuscript. All the authors have consented the authorship, reviewed, and approved the paper for publication.

## Competing interests

The University of Massachusetts Chan Medical School has patented the JAK1 siRNA sequences (Patent publication numbers: WO2022271666A1; Oligonucleotides for IFN-γ signaling pathway

modulation) and the docosanoic acid conjugate (CA3174068A1; Conjugated oligonucleotides for tissue-specific delivery) for the use with therapeutic oligonucleotides, and the outlined technologies in this article has been licensed to Aldena Therapeutics for clinical development. A.K. discloses ownership of stocks in RXi Pharmaceuticals and Advirna; is a founder of Atalanta Therapeutics and Comanche Biopharma; serves on the Scientific Advisory Board of Aldena Therapeutics, Prime Medicine, and Alltrna. J.E.H. holds equity in Rheos Medicines and TeVido BioDevices; is a founder with equity of Villaris Therapeutics, Aldena Therapeutics, NIRA Biosciences, Vimela Therapeutics, and Klirna Therapeutics; has served as a consultant for Pfizer, Sanofi Genzyme, Incyte, Sun Pharmaceuticals, LEO Pharma, Dermavant, Temprian Therapeutics, AbbVie, Janssen, Almirall, Methuselah Health, Pandion, AnaptysBio, Avita, Aclaris Therapeutics, The Expert Institute, BiologicsMD, Boston Pharma, Sonoma Biotherapeutics, Two Biotech, Admirx, Frazier Management, 3rd Rock Ventrures, Gogen Therapeutics, Granular Therapeutics, Platelet Biogenesis, BridgeBio, Merck, Matchpoint Therapeutics, and Klirna; has served as an investigator for Pfizer, Sanofi Genzyme, Incyte, Sun Pharmaceuticals, LEO Pharma, Dermavant, Aclaris Therapeutics, GSK, Celgene, Dermira, and EMD Serono. C.B.P. and C. Blanchard are employed executives of Aldena Therapeutics. The remaining authors declare no competing interests.
