## [Peer Review File · Nature Communications]

Reviewers' Comments:

Reviewer #1:

Remarks to the Author:

Tang et al. describe development and characterization of therapeutic JAK1 siRNA for in vivo use. Overall this is a very well designed study and clearly written paper and I applaud the authors for their innovation. Below are comments that may help the authors further improve the manuscript and project

Major

1. systemic exposure is documented. stated that the molecule is safe based on limited serum chemistries and citing prior literature on safety of similar RNA drugs. the discussion of safety appears a bit premature:

- Based on EDF2 panels c and d the level of the si3033 in tissue is surprisingly extensive. I think it would be important to evaluate histology of organs where uptake is so extensive- is it normal? JAK1 is constitutively expressed by non immune cells and important for many functions.

- Also although serum chemistries are measured and appear unaffected, what about hematologic parameters? systemic exposure to JAK1 inhib could affect cell counts, etc in blood . are cd8 T cells reduced systemically (for example)?

In thinking about how this might be used in the clinic, it would be helpful for the authors to also discuss possible limitations. thinking about patients with even just 1% BSA vitiligo multiple injections would be needed to cover just that 1% BSA and would be needed over time as well, problem compounded in folks with more extensive involvement. although this is framed as improvement over ruxolitinib due to more specific inhibition JAK1 not JAK1/2, it is not really demonstrated that this is superior to ruxolitinib either in terms of safety or efficacy and there seems to be a clear downside to the means by which it needs to be administered compared to a cream. can the authors address this as well briefly in the discussion? Agree with authors that diseases like alopecia areata where deep delivery is important could also be interesting to pursue.

Minor

- measurement of JAK1 protein in skin tissue - is it reduced? Assuming it is based on all of the other data but would be useful to know to what extent this happens.

- are number of PMEL cells reduced systemically (in addition to treated footpad)? stated that si3033 reduces migration of cells into the skin- do you really know its migration as opposed to survival, local proliferation, etc?

Reviewer #2:

Remarks to the Author:

The paper 'Rational design of a JAK1-selective siRNA inhibitor for the modulation of autoimmunity in the skin' provides in-depth insight into opportunities to selectively and effectively inhibit JAK1 on location, providing a safer means of tempering cutaneous autoimmune responses, and more specifically: vitiligo. JAK inhibitors have already revolutionized the field. Meanwhile, the proposed method of inhibition can overcome some of the main limitations to JAK inhibition, namely its limited specificity for the JAK isoform critical to disease development and its unintended reach beyond treatment areas that require treatment. The impact for vitiligo patients as well as patients with other inflammatory conditions, or with vitiligo secondary to their melanoma treatment cannot easily be overstated. The work is very comprehensive and important, with a few points to be addressed:

- In practice, patients will come in for treatment while experiencing active disease. However, the paper is focused on preventing depigmentation in the absence of inflammation or autoimmune activity. JAK Inhibition prior to vitiligo induction is not as clinically meaningful as treating mice in the effector phase of disease and exhibiting active depigmentation. Certainly the % inhibition required before disease becomes active cannot be directly compared to the % inhibition needed to

halt ongoing disease (Fig 3f, EFig4). It will be important to include experiments, at least an in vitro experiment, using pre-stimulation with IFN- γ to emulate active disease and treat with 3033 when activity is in full swing.

- In Fig. 4f, the quantification of downstream events related to IFN- γ signaling is meticulously tested; it will also be important to learn of/comment upon the specificity of such effects and understand if non-IFN- γ signaling is differentially affected as well, especially IL-10 related downstream signaling.

- Fig. 3h needs quantification and/or showing all mice as the difference in depigmentation between both feet shown for one mouse alone is not convincing.

- A limitation that may remain is the need for continuous treatment. Though effective delivery through cholesterol or docosanoic acid conjugation is a plus, the authors might comment on additional measures that could allow for prolonged efficacy.

- Testing the ability of the lead compound to inhibit IFN- γ downstream signaling in a reporter cell line: does the reporter cell line exhibit functional expression of JAKs other than JAK1 to test the redundancy of JAK1 functions? Discuss: what is the evidence that other JAKs do not contribute to IFN- γ signaling?

- The authors should discuss/consider the meaning of inhibiting JAK1 in C57BL/6 mouse skin without inflammation or autoimmunity; How does it compare to inhibiting JAK1 by injection into inflamed skin, and how does it compare to ruxolitinib mediated inhibition?

- Upon systemic injection of si3033, what is the uptake profile of the drug by leukocytes? (See Fig. 3c). Was this performed in mice with inflammation or without?

- There is concern about the uptake efficiency by keratinocytes, the cell type which appears to be important for generating mediators of T cell recruitment to the skin. Meanwhile the role of fibroblasts has not been accentuated in research to date. It can be helpful to discuss alternate means of introducing si3033 to the skin that could favor keratinocyte uptake and potentially have additional benefits for treatment.

- The improved activity of 3033 versus ruxolitinib appears directly related to the concentration for either treatment (Fig 2 i,j); the authors might also comment on the differential stabilities of either drug/treatment.

John E. Harris, M.D., Ph.D.
Professor and Chair, Department of Dermatology
University of Massachusetts Chan Medical School
364 Plantation St, Worcester, MA 01605
john.harris@umass.edu

Anastasia Khvorova, Ph.D.
Professor, RNA Therapeutics Institute
University of Massachusetts Chan Medical School
368 Plantation St, Worcester, MA 01605
anastasia.khvorova@umassmed.edu

Julia Alterman, Ph.D.
Assistant Professor, RNA Therapeutics Institute
University of Massachusetts Chan Medical School
368 Plantation St, Worcester, MA 01605
julia.alterman@umassmed.edu

July 30, 2023

Dear reviewers,

We would like to thank you for your thoughtful and constructive comments on our manuscript. These comments have greatly helped us strengthen the quality of this work. To address your questions, we have carried out additional studies and incorporated new data or corrections to reflect most of the suggestions in your comments. Please find the point-by-point response as below.

Reviewer #1:

1. “Tang et al. describe development and characterization of therapeutic JAK1 siRNA for in vivo use. Overall this is a very well designed study and clearly written paper and I applaud the authors for their innovation. Below are comments that may help the authors further improve the manuscript and project”

We really appreciate this highly positive feedback to our work.

Major

2. “1. systemic exposure is documented. stated that the molecule is safe based on limited serum chemistries and citing prior literature on safety of similar RNA drugs. the discussion of safety appears a bit premature:” “- Also although serum chemistries are measured and appear unaffected, what about hematologic parameters? systemic exposure to JAK1 inhib could affect cell counts, etc in blood . are cd8 T cells reduced systemically (for example)?”

Generally, the safety of a therapy is dependent on multiple factors including target selection, chemical toxicity, administrative route, and dosage, etc. JAK1 is a clinically validated target for treating many immune disorders. We evaluated the chemical configurations, administrative routes, and dosing regimen of si3033, aiming to advance the use of this drug for locally modulating inflammation in skin conditions (e.g., alopecia areata and vitiligo) with depth-controlled delivery methods (i.e., transdermal or intradermal). For changes in hematologic parameters, we systemically injected mice with 20 mg/kg of si3033 and measured the complete blood counts at 24 h and 72 h post treatment. The data shown below indicates in general a good safety profile of si3033 except a transient decrease of platelet counts at 24 h; this side effect was recovered at 72 h. This data has now been included to the revised manuscript and being discussed in detail in the results section. The intended human use of si3033 is through local administration (e.g., intradermal,

topical formulation, or other transdermal delivery methods). We hypothesized that administration of hydrophobically modified si3033 into thick human skin would allow large fraction of the compounds being retained locally, thus reducing systemic exposure. This hypothesis has been demonstrated in our latest preclinical pharmacokinetic studies

(i.e., intradermal injection of si3033) in porcine skin *in vivo*, pig is considered as the most accurate comparative model of human for dermatological studies due to its similarity in structural skin characteristics. The intended maximum dose of si3033 for clinical trial is predicted to be as low as 0.43 mg/kg. This dose is calculated from the repeated injection of 50 μL (400 μM) si3033 in an 80 cm^2 surface area at every 1 cm^2 . The cumulative total dose injected is 21 mg per individual assuming a body weight of 50 kg. Non-GLP pilot studies with intradermal delivery of si3033 were conducted in five 4-month old Göttingen minipigs with 80 repeated dosing of si3033 (total 21 mg). There were no dose-related findings in the heart, liver, kidneys, or spleen. There were no dose-related systemic toxicity findings from hematology or clinical laboratories. Platelet levels were measured at 24h, 48h, and then weekly through Day 28 as shown in the figure to the right, indicating good safety profile of si3033 for the intended dosing regimen in clinic. The detailed data will be published in a separate manuscript along with ongoing efficacy and toxicity studies.

For the question regarding the systemic suppression of CD8^+ T cells after si3033 treatment, there was no sign of significant reduction of CD8^+ T cells. This question will be addressed along with your similar question in the comment 6 of this response letter with supporting data.

3. *“- Based on EDF2 panels c and d the level of the si3033 in tissue is surprisingly extensive. I think it would be important to evaluate histology of organs where uptake is so extensive- is it normal? JAK1 is constitutively expressed by non immune cells and important for many functions.”*

Thank you for your suggestion. We agree that histology evaluation in addition to the blood diagnostics is important for assessing the safety of si3033 in organs that had extensive siRNA accumulation. We thus conducted histology studies in tissues (i.e., local skin, liver, spleen, kidney) with high si3033 accumulation. The H&E stained samples (by Morphology Research Core at UMass Chan Medical School) from PBS and si3033 treated mice (20 mg/kg, 7 days post S.C. Injection) were blinded and independently assessed (by histopathologist in the Department of Pathology at UMass Chan Medical School). There were no findings of histological changes (e.g., morphological, architectural, and colorimetric features) in the cells and tissues between the two groups. The results and histological images were included in the revised manuscript (Extended fig 2e) as shown below.

4. *“In thinking about how this might be used in the clinic, it would be helpful for the authors to also discuss possible limitations. thinking about patients with even just 1% BSA vitiligo multiple injections would be needed to cover just that 1% BSA and would be needed over time as well, problem compounded in folks with more extensive involvement. although this is framed as improvement over ruxolitinib due to more specific inhibition JAK1 not JAK1/2, it is not really demonstrated that this is superior to ruxolitinib either in terms of safety or efficacy and there seems to be a clear downside to the means by which it needs to be administered compared to a cream. can the authors address this as well briefly in the discussion? Agree with authors that diseases like alopecia areata where deep delivery is important could also be interesting to pursue.”*

We fully agree that for skin conditions that large body surface is involved the proposed local injection is not ideal in the clinic. To address this concern, we would like to point out the latest efforts in the field including our group who are actively working on developing novel approaches to functionally deliver large oligonucleotide-based therapeutics into the skin.

We have been collaborating with Dr. Mark Prausnitz group at Georgia Tech to apply a patient-friendly micro-STAR particle technology for depth-controlled delivery of nucleic acid therapies into human skin. These engineered micro-STAR particles can facilitate topical drugs to penetrate stratum corneum by creating transient non-invasive pores. This technology has been evaluated in human subject for its tolerability. In addition, there are other advances in the field for skin delivery of oligonucleotides. For example, Samir Mitragotri group at Harvard previously described functional skin delivery of siRNA using topical ionic liquid formulations. These technologies are potentiating topical use of siRNAs for skin conditions with extensive skin area involvement and are described in detail in the below publications. We believe with further technology exploration, topical delivery of siRNA-based drugs is promising and will be available for the treatment of a number of skin indications upon the accomplishment of clinical validation. As suggested, we have now briefly discussed as below the limitation of our approach and possible solutions in the discussion section of the manuscript to address this concern.

“For skin conditions that involve large area of body surface that intradermal injection of siRNA is not straightforward, novel depth-controlled transdermal delivery methods would be beneficial and is an area being actively explored (e.g., microparticle and novel skin-penetrating formulations).”

- (1). R. Tadros, A. Romanyuk, I. C. Miller, A. Santiago, R. K. Noel, L. O’Farrell, G. A. Kwong, M. R. A. R. Tadros, A. Romanyuk, I. C. Miller, A. Santiago, R. K. Noel, L. O’Farrell, G. A. Kwong, M. R. Prausnitz, STAR particles for enhanced topical drug and vaccine delivery. *Nature Medicine* 26, 341–347 (2020).
- (2). Y. Kim, J. H. Jung, A. R. Tadros, M. R. Prausnitz, Tolerability, acceptability, and reproducibility of topical STAR particles in human subjects. *Bioengineering & Translational Medicine* 8, e10524 (2023).
- (3). Mandal, N. Kumbhojkar, C. Reilly, V. Dharamdasani, A. Ukidve, D. E. Ingber, S. Mitragotri, Treatment of psoriasis with NFKBIZ siRNA using topical ionic liquid formulations. *Science Advances* 6, eabb6049 (2020).

(4). V. Dharamdasani, A. Mandal, Q. M. Qi, I. Suzuki, M. V. L. B. Bentley, S. Mitragotri, Topical delivery of siRNA into skin using ionic liquids. *Journal of Controlled Release* 323, 475–482 (2020).

“Minor

5. measurement of JAK1 protein in skin tissue - is it reduced? Assuming it is based on all of the other data but would be useful to know to what extent this happens.”

The reduced production of JAK1-dependent downstream chemokines CXCL9 and CXCL10 in skin tissues clearly indicate less available JAK1 protein for signaling transduction after si3033 treatment. To provide direct evidence as suggested, we measured the level of JAK1 protein 2 weeks post injection of si3033 or non-targeting control siRNA (identical treatment regimen as in Fig 3d and 3e) in mouse footpad skins using Wes capillary assay. We found that JAK1 protein silencing level (~35% reduction) was well correlated to the *Jak1* mRNA and its downstream chemokine reduction levels in si3033 treated footpad skin. This data as shown in the right graph has now been added to the manuscript as Extended Fig 3c.

6. “- are number of PMEL cells reduced systemically (in addition to treated footpad)? stated that si3033 reduces migration of cells into the skin- do you really know its migration as opposed to survival, local proliferation, etc?”

The hydrophobically conjugated JAK1 siRNA was designed to enhance its local retention in the skin. The intended use for human skin by superficial transdermal delivery or intradermal delivery should greatly reduce its systemic exposure; thus JAK1-related systemic immunosuppression is predicted to be limited. This hypothesis was supported by our latest unpublished data in pig models *in vivo*. To address the question whether si3033 may lead to systemic inhibition of PMEL CD8⁺ T cell proliferation, we have previously evaluated the systemic administration of very high dose (S.C., 20 mg/kg weekly for 5 weeks) of si3033 in mice and measured the PMEL CD8⁺ T cells in spleen and lymph nodes. The below data showed systemic exposure did not result in significant impact on PMEL numbers even at the exaggerated dose.

Whether the reduced PMEL CD8⁺ T cells in the treated footpad was due to less migration as opposed to survival, or local proliferation, etc.? This is a very good point and it is possible, as immune responses at the local environment could be complicated given the fact that JAK1 is also involved in signaling pathways that responsible for T cell activation and proliferation. We believe the observed phenotype in this work was more likely due to less PMEL migration thus led to reduced T cell-mediated cytotoxicity to melanocytes. This is because we have learned that systemic injection of high dose of si3033 did not result in significant T cell population decrease in spleen and lymph nodes, whereas the treated footpad had significant T cell numbers reduction; thus si3033 may have less impact on PMEL T cell proliferation and survival than on the recruitment of T cells to the site of autoimmunity.

Reviewer #2:

7. “The paper ‘Rational design of a JAK1-selective siRNA inhibitor for the modulation of autoimmunity in the skin’ provides in-depth insight into opportunities to selectively and effectively inhibit JAK1 on location, providing a safer means of tempering cutaneous autoimmune responses, and more specifically: vitiligo. JAK inhibitors have already revolutionized the field. Meanwhile, the proposed method of inhibition can overcome some of the main limitations to JAK inhibition, namely its limited specificity for the JAK isoform critical to disease development and its unintended reach beyond treatment areas that require treatment. The impact for vitiligo patients as well as patients with other inflammatory conditions, or with vitiligo secondary to their melanoma treatment cannot easily be overstated. The work is very comprehensive and important, with a few points to be addressed.”

We would like to thank you for your opinion on supporting the significance of this project. To address your questions, we have provided detailed response.

8. “- In practice, patients will come in for treatment while experiencing active disease. However, the paper is focused on preventing depigmentation in the absence of inflammation or autoimmune activity. JAK Inhibition prior to vitiligo induction is not as clinically meaningful as treating mice in the effector phase of disease and exhibiting active depigmentation. Certainly the % inhibition required before disease becomes active cannot be directly compared to the % inhibition needed to halt ongoing disease (Fig 3f, EFig4). It will be important to include experiments, at least an in vitro experiment, using pre-stimulation with IFN- γ to emulate active disease and treat with 3033 when activity is in full swing.”

We agree that therapeutic intervention for active disease is more meaningful from clinical perspective. This had been tried and it was challenging to obtain consistent phenotypic readouts (reversed depigmentation or repigmentation) in the treated footpad of our vitiligo model. The mouse model mechanistically mimics the CD8⁺ T cell-mediated autoimmunity of human vitiligo; however, has accelerated inflammation and disease progression. We found that preventive administration of siRNA is required for phenotypic efficacy (prevention of depigmentation), this was because optimal silencing of JAK1 was not achieved when the disease was initiated (adoptive transfer and activate CD8⁺ T cells). Based on our *in vivo* results (Fig. 3d), it takes approximately 2 weeks to efficiently silence JAK1 in the footpad skin. During this time period, CD8⁺ T cell-mediated cytotoxicity on melanocytes progresses aggressively that an on-time or high level of JAK1 silencing might be required, which is technically hard to achieve in the model. As the pharmacodynamic and pharmacokinetic properties of siRNA in mouse skin dramatically differ from human skin, an appropriate animal model would provide better insights into the therapeutic potential of the siRNA, as what we are currently doing in pigs.

As suggested by your comments, to better understand the therapeutic potential of the siRNA after disease progression, we carried out *in vitro* studies in mouse keratinocytes to mimic the therapeutic modulation of active inflammation using 3033. The IFN- γ signaling was stimulated in mouse PAM212 cell line for 6 hours and we confirmed there was a significant upregulation of the downstream chemokine genes *Cxcl9* and *Cxcl10* at this time point. The cells were then treated with control siRNA (NTC) and 3033, we were able to observe efficient reduction of the inflammation over 4 days by 3033 compared to NTC as shown in below figures. The *in vivo* treatment of vitiligo mice after disease progression has multiple layer of complexity with the involvement of many factors, but we are actively optimizing the potency of the siRNAs and experimental conditions for better understanding the dosing regimen to inform the clinical development of si3033.

Revision Figure. Mouse PAM212 keratinocyte cell line was used in this experiment with 3000 cells seeding density per well in a standard 96-well plate. 10 ng/mL of recombinant murine IFN- γ was used to stimulate the IFN- γ signaling; and 10 ng/mL of TNF- α was added to enhance the signaling activation level. The expression of mouse Cxcl9 and Cxcl10 was measured using QuantiGene 2.0 assays (n=7). Basic expression level of Cxcl10 gene was commonly observed in a number of cell lines, which explains the elevation of Cxcl10 mRNA level at later time points (i.e., 72 and 96 hours) of unstimulated cells in this study due to the proliferation of cells. si3033 efficiently suppressed the activated inflammatory signaling and also depleted the background level of Cxcl10 expression.

9. “- In Fig. 4f, the quantification of downstream events related to IFN- γ signaling is meticulously tested; it will also be important to learn of/comment upon the specificity of such effects and understand if non-IFN- γ signaling is differentially affected as well, especially IL-10 related downstream signaling.”

The upregulation of CXCL9/10/11 chemokines was specific to IFN- γ signaling activation in the tested *ex vivo* skin culture model upon the addition of human IFN- γ . Although JAK1 is involved in signaling transduction of IL10, we checked our RNAseq data and did not observe any IL10 signaling activation related events (all sequencing data generated from this work will be deposited and shared in public database).

10. “- Fig. 3h needs quantification and/or showing all mice as the difference in depigmentation between both feet shown for one mouse alone is not convincing.”

We show below, in panel (a), the footpad images of all treated mice reflecting the quantification of PMEL CD8⁺ T cells in the si3033 injected vs non-targeting control siRNA (siNTC) treated footpads (Fig.3g). The prevention of skin infiltration of CD8⁺ T cell due to reduced inflammation from JAK1 silencing was clear despite of there was variability in disease progression (i.e., epidermal depigmentation) of a few mice. Additionally, in an independent earlier study of screening the efficacy of multiple siRNA 3033 chemical scaffolds; we observed, in panel (b), similar phenotypic efficacy using DCA-conjugated siRNA (i.e., si3033 in this manuscript); unfortunately, we did not analyze the PMEL CD8⁺ T cells in those skin samples due to the overwhelmingly large sample size in the screening study.

We are working on establishing objective quantification methods to measure the skin pigment intensity in our vitiligo mouse model, which including: (a) quantifying melanin level using ELISA assay; (b) applying melanin sensor electronic probe (melanometer) to quantify pigment intensity. Objective quantification of the depigmentation level is important for supporting the PMEL CD8⁺ T cell infiltration data; unfortunately, we experienced technical limitation during the course of this study; thus were not able to collect useful data. We would like to thank you for the suggestion, and we will continue to improve the objective quantification approaches.

11. “- A limitation that may remain is the need for continuous treatment. Though effective delivery through cholesterol or docosanoic acid conjugation is a plus, the authors might comment on additional measures that could allow for prolonged efficacy.”

Thank for you this suggestion. Indeed, we have recently developed a novel biologically compatible siRNA backbone modification (i.e., extended nucleic acid: exNA) that significantly enhances *in vivo* duration of effect and potency of therapeutic oligonucleotide due to improved metabolic stability. Multiple modifications of 5' phosphate of antisense strand are also being developed in the lab. Due to limited space, we provided a short prospect of this topic as below in the discussion section of the revised manuscript; some of the work is currently under revision for publication and will be soon available for scientific community.

“Additionally, biologically compatible chemical modifications of 3' phosphate backbone and 5' phosphate of antisense strand should provide extra metabolic stability of siRNA; thus allowing for prolonged efficacy and less frequent dosing in the clinic.”

12. *“- Testing the ability of the lead compound to inhibit IFN- γ downstream signaling in a reporter cell line: does the reporter cell line exhibit functional expression of JAKs other than JAK1 to test the redundancy of JAK1 functions? Discuss: what is the evidence that other JAKs do not contribute to IFN- γ signaling?”*

IFN- γ signaling is well-characterized and documented, the signaling transduction is solely dependent on JAK1 and JAK2 subtypes. The other two JAK family enzymes JAK3 and TYK2 are not involved in IFN- γ signaling pathway. We believe JAK1 is not redundant in this reporter cell line as our data showed that selective silencing of JAK1 (Fig.2j) was sufficient to block IFN- γ signaling transduction.

13. *“- The authors should discuss/consider the meaning of inhibiting JAK1 in C57BL/6 mouse skin without inflammation or autoimmunity; How does it compare to inhibiting JAK1 by injection into inflamed skin, and how does it compare to ruxolitinib mediated inhibition?”*

Thank you for the suggestion and this question has also been raised by reviewer 1 in comment 8. We have partially addressed the concern by demonstrating the effect of siRNA modulating preexisting inflammation in an *in vitro* system with data shown as in the above response to comment 8. Injecting siRNA before inflammation is not ideal but represents a viable path to understand the therapeutic potential of modulating autoimmunity by selectively silencing JAK1. Therapeutically treat vitiligo mice with accelerated autoimmunity and slow repigmentation process is technically challenging as discussed above in detail.

Benchmarking ruxolitinib to JAK1 siRNA *in vivo* might not be necessary as the route of administration and action of mechanism of the two therapeutic modalities are different from several perspectives, which requires well-controlled and optimized experimental conditions that are technically challenging to perform. We provided our prospects on how siRNA therapeutics might differ from the current therapeutic paradigm of using small molecule topical JAK inhibitors as below.

“Topical use of small molecule ruxolitinib is patient friendly, however, requires twice daily application to the affected skin areas of only up to 10% body surface due to safety concerns, satisfactory patient response may require treatment for more than 24 weeks. siRNAs differ from small molecules in action of mechanism. Upon cellular internalization, entrapment of metabolically stabilized siRNAs in lysosomal and endosomal compartments generates an intracellular depot of the drug with slow release, which offers improved potency and duration of effect. For skin conditions that involve large area of body surface that intradermal injection of siRNA is not straightforward, novel depth-controlled transdermal delivery methods would be beneficial and is an area being actively explored (e.g., microparticle and novel skin-penetrating formulations).”

14. *“- Upon systemic injection of si3033, what is the uptake profile of the drug by leukocytes? (See Fig. 3c). Was this performed in mice with inflammation or without?”*

In this study, we only characterized the uptake profile of DCA-siRNA (si3033) in cell types of human skin after intradermal injection; and we found in general the uptake si3033 in leukocytes was less than in resident skin cell types. In Fig. 3c, the study was performed in mice without inflammation, we did not characterize the leukocytes uptake of si3033 systemically in these mice. However, we have previously evaluated certain leukocyte uptake profile of DCA-conjugated siRNA in mice showing their relative uptake efficiency and the data are shown as below. As bioconjugates but not sequence dictate the cell type uptake profile of siRNAs, we would expect similar results for si3033.

15. “- There is concern about the uptake efficiency by keratinocytes, the cell type which appears to be important for generating mediators of T cell recruitment to the skin. Meanwhile the role of fibroblasts has not been accentuated in research to date. It can be helpful to discuss alternate means of introducing si3033 to the skin that could favor keratinocyte uptake and potentially have additional benefits for treatment.”

This is a very important point as achieving preferential cell type uptake for RNA therapeutics is a major research focus of the field. It is beneficial to improve the delivery of therapeutic RNAs into the cell types that are relevant to the disease pathology; and free bystander cell types to avoid undesired side effects. We have previously demonstrated that the inhibition of IFN- γ signaling in epidermal keratinocytes is sufficient to prevent skin depigmentation in our vitiligo mouse model, this might also be true for certain epidermal inflammatory skin diseases. Thus, improving keratinocyte delivery of siRNA using alternative strategies might be helpful. To provide prospect of this topic, we have added into the discussion section of the manuscript with below content:

“Our RNAseq data highlight the potential of si3033 to modulate a broad range of inflammatory responses upon IFN- γ stimulation in human skin; additional work will be required to dissect the complexity of these biological events. A clearer understanding of these events, especially under pathological conditions in a cell-type specific resolution could better inform the therapeutic intervention of many diseases driven by dysregulated IFN- γ signaling. Correspondingly, more efficient delivery of siRNA drugs into skin cell types that drive disease progression would be advantageous, this may be achieved through antibody conjugates against cell type-specific markers or novel small RNA trafficking mechanisms.”

16. “- The improved activity of 3033 versus ruxolitinib appears directly related to the concentration for either treatment (Fig 2 i,j); the authors might also comment on the differential stabilities of either drug/treatment.”

si3033 and small molecule ruxolitinib are different drug modalities that work in distinct action of mechanisms. Ruxolitinib directly binds to the kinase domain of JAK1 and JAK2 to inhibit their activity. In comparison, si3033 is internalized into cells through endocytosis and are mostly entrapped in lysosomal

and endosomal compartments, generating an intracellular depot of the drug with slow release. In addition, the inhibition of JAK protein by ruxolitinib is stoichiometric, whereas the cleavage of mRNAs by siRNA is catalytic. We compared in a cellular assay the activity of 3033 to ruxolitinib using identical concentration (i.e., 1.5 μM) as described in Fig. 2c, 3033 clearly showed improved activity in reducing JAK1-dependent IFN- γ signaling activation. We fully agree that the stability of drug compounds may be attributed to their potency, this could be one reason why chemically stabilized si3033 showed improved activity over ruxolitinib. We briefly described this topic as integrated into the response to comment 13.

We hope our responses have fully addressed the questions in your comments. Finally, we would like to thank you again for your invaluable opinion on this work and we look forward to addressing any additional questions you may have.

Sincerely,

John E. Harris, Anastasia Khvorova, and Julia Alterman

Reviewers' Comments:

Reviewer #1:

Remarks to the Author:

All of my concerns have been address. Very impressive.

Reviewer #2:

Remarks to the Author:

The manuscript has very much improved. Referencing the prior responses to numbered questions, a few (partial) questions remain:

9. The upregulation of CXCL9/10/11 chemokines was specific to IFN- γ signaling activation in the tested ex vivo skin culture model upon the addition of human IFN- γ . Although JAK1 is involved in signaling transduction of IL10, we checked our RNAseq data and did not observe any IL10 signaling activation related events (all sequencing data generated from this work will be deposited and shared in public database).

Question: Can the authors comment on non-IFN-gamma related signaling impacted by the proposed siRNA? Even if we might learn this once the sequencing data are published in full, the authors might highlight some meaningful non-IFN-gamma related transcripts altered by the proposed treatment, or otherwise mention that the phenotype of treated mice appears to be the same or similar as that of mice wherein IFN-gamma is knocked down/depleted?

10. We show below, in panel (a), the footpad images of all treated mice reflecting the quantification of PMEL CD8+ T cells in the si3033 injected vs non-targeting control siRNA (siNTC) treated footpads (Fig.3g). The prevention of skin infiltration of CD8+ T cell due to reduced inflammation from JAK1 silencing was clear despite of there was variability in disease progression (i.e., epidermal depigmentation) of a few mice. Additionally, in an independent earlier study of screening the efficacy of multiple siRNA 3033 chemical scaffolds; we observed, in panel (b), similar phenotypic efficacy using DCA-conjugated siRNA (i.e., si3033 in this manuscript); unfortunately, we did not analyze the PMEL CD8+ T cells in those skin samples due to the overwhelmingly large sample size in the screening study. We are working on establishing objective quantification methods to measure the skin pigment intensity in our vitiligo mouse model, which including: (a) quantifying melanin level using ELISA assay; (b) applying melanin sensor electronic probe (melanometer) to quantify pigment intensity. Objective quantification of the depigmentation level is important for supporting the PMEL CD8+ T cell infiltration data; unfortunately, we experienced technical limitation during the course of this study; thus were not able to collect useful data. We would like to thank you for the suggestion, and we will continue to improve the objective quantification approaches.

Question: Looking at the full dataset, the difference in depigmentation between both feet may be significant, but that is difficult to judge with the naked eye. Quantifying the outcomes through image analysis can make this figure much more convincing

12. IFN- γ signaling is well-characterized and documented, the signaling transduction is solely dependent on JAK1 and JAK2 subtypes. The other two JAK family enzymes JAK3 and TYK2 are not involved in IFN- γ signaling pathway. We believe JAK1 is not redundant in this reporter cell line as our data showed that selective silencing of JAK1 (Fig.2j) was sufficient to block IFN- γ signaling transduction.

Question: If the reporter cell line indeed expresses JAK2 and this is mentioned in the paper together with the last sentence (We believe that transduction), this response will be clear and sufficient.

John E. Harris, M.D., Ph.D.
Professor and Chair, Department of Dermatology
University of Massachusetts Chan Medical School
364 Plantation St, Worcester, MA 01605
john.harris@umass.edu

Anastasia Khvorova, Ph.D.
Professor, RNA Therapeutics Institute
University of Massachusetts Chan Medical School
368 Plantation St, Worcester, MA 01605
anastasia.khvorova@umassmed.edu

Julia Alterman, Ph.D.
Assistant Professor, RNA Therapeutics Institute
University of Massachusetts Chan Medical School
368 Plantation St, Worcester, MA 01605
julia.alterman@umassmed.edu

August 27, 2023

Dear reviewers,

We highly appreciate your constructive feedback to our revised manuscript. Based on your comments, we have performed additional analysis of the data and rephrased the manuscript to reflect the corrections. Please find below our point-by-point responses.

Reviewer #1: All questions have been addressed.

Reviewer #2:

“The manuscript has very much improved. Referencing the prior responses to numbered questions, a few (partial) questions remain:”

9. The upregulation of CXCL9/10/11 chemokines was specific to IFN- γ signaling activation in the tested ex vivo skin culture model upon the addition of human IFN- γ . Although JAK1 is involved in signaling transduction of IL10, we checked our RNAseq data and did not observe any IL10 signaling activation related events (all sequencing data generated from this work will be deposited and shared in public database).

Question: *“Can the authors comment on non-IFN-gamma related signaling impacted by the proposed siRNA? Even if we might learn this once the sequencing data are published in full, the authors might highlight some meaningful non-IFN-gamma related transcripts altered by the proposed treatment, or otherwise mention that the phenotype of treated mice appears to be the same or similar as that of mice wherein IFN-gamma is knocked down/depleted?”*

As JAK1 is involved in mediating multiple inflammatory signaling pathways, it might be unavoidable that silencing of JAK1 will impact non-IFN-gamma related signaling. The secondary responses upon IFN-gamma pathway activation might also be affected by JAK1 inhibition. The underlying biology whether other related signaling pathways were attributed to the phenotype of treated mice is challenging to deconvolute in this case, however, we have previously show similar therapeutic effects by directly targeting the IFN-gamma cytokine using antibodies as shown in the below reference, and IFN-gamma receptor knockout host mice were fully protected from developing diseases (data will be published soon).

Y.-C. Hsueh, Y. Wang, R. L. Riding, D. E. Catalano, Y.-J. Lu, J. M. Richmond, D. L. Siegel, M. Rusckowski, J. R. Stanley, J. E. Harris, A Keratinocyte-Tethered Biologic Enables Location-Precise Treatment in Mouse Vitiligo. *Journal of Investigative Dermatology*, S0022202X22016475 (2022).

10. We show below, in panel (a), the footpad images of all treated mice reflecting the quantification of PMEL CD8+ T cells in the si3033 injected vs non-targeting control siRNA (siNTC) treated footpads (Fig.3g). The prevention of skin infiltration of CD8+ T cell due to reduced inflammation from JAK1 silencing was clear despite of there was variability in disease progression (i.e., epidermal depigmentation) of a few mice. Additionally, in an independent earlier study of screening the efficacy of multiple siRNA 3033 chemical scaffolds; we observed, in panel (b), similar phenotypic efficacy using DCA-conjugated siRNA (i.e., si3033 in this manuscript); unfortunately, we did not analyze the PMEL CD8+ T cells in those skin samples due to the overwhelmingly large sample size in the screening study. We are working on establishing objective quantification methods to measure the skin pigment intensity in our vitiligo mouse model, which including: (a) quantifying melanin level using ELISA assay; (b) applying melanin sensor electronic probe (melanometer) to quantify pigment intensity. Objective quantification of the depigmentation level is important for supporting the PMEL CD8+ T cell infiltration data; unfortunately, we experienced technical limitation during the course of this study; thus were not able to collect useful data. We would like to thank you for the suggestion, and we will continue to improve the objective quantification approaches.

Question: “Looking at the full dataset, the difference in depigmentation between both feet may be significant, but that is difficult to judge with the naked eye. Quantifying the outcomes through image analysis can make this figure much more convincing.”

Thank you for your suggestion. We have now analyzed the images and quantified the distribution profiles of pigment intensity of all footpads using ImageJ software. The data (attached to this file) now are presented as the Extended Data Fig.5 in the revised manuscript.

12. IFN- γ signaling is well-characterized and documented, the signaling transduction is solely dependent on JAK1 and JAK2 subtypes. The other two JAK family enzymes JAK3 and TYK2 are not involved in IFN- γ signaling pathway. We believe JAK1 is not redundant in this reporter cell line as our data showed that selective silencing of JAK1 (Fig.2j) was sufficient to block IFN- γ signaling transduction.

Question: “If the reporter cell line indeed expresses JAK2 and this is mentioned in the paper together with the last sentence (We believe that transduction), this response will be clear and sufficient.”

We have clarified the statement in the revised manuscript as shown below.

As si3033 selectively silences JAK1 mRNA and does not impact the expression of JAK2 (Fig. 2j), this result suggests that selectively targeting JAK1 may be sufficient to inhibit JAK1/2-mediated inflammatory pathways. Our findings support a selective JAK inhibition strategy for reducing the undesired effects of inhibiting multiple JAKs without compromising efficacy.

Finally, we would like to thank you again for all the comments that greatly helped us the enhance the quality of this work. We look forward to addressing any additional questions you may have.

Sincerely,

John E. Harris, Anastasia Khvorova, and Julia Alterman

Extended Data Fig.5 | si3033 prevents skin pigmentation in an autoreactive CD8⁺ T cell-mediated vitiligo mouse model. a. Quantification of pigment intensity of the treated footpads ($n=12$; right: siNTC, left si3033). Pigment distribution profiles were analyzed using histogram analysis in Image J Fiji software and presented as the pixel numbers of each intensity value in the range of 0-255 (0: dark value, and 255: white value). **b.** The phenotype of si3033 treatment was reproduced in an independent experiment ($n=10$). As the disease progression (depigmentation) has inter-individual variability, the pigment distribution profile of si3033 treated footpad was visualized by aligning to the profile the siNTC treated footpad in each mouse.

Reviewers' Comments:

Reviewer #2:

Remarks to the Author:

The authors have provided satisfactory responses to the remaining questions, and this exciting work is ready to be shared with the research community.